# Extrinsic and intrinsic dynamics in movement intermittency

**Damar Susilaradeya[†], Wei Xu, Thomas M Hall, Ferran Galán[‡], Kai Alter, Andrew Jackson***

Institute of Neuroscience, Faculty of Medical Sciences, Newcastle University, Newcastle, United Kingdom

**Abstract** What determines how we move in the world? Motor neuroscience often focusses either on intrinsic rhythmical properties of motor circuits or extrinsic sensorimotor feedback loops. Here we show that the interplay of both intrinsic and extrinsic dynamics is required to explain the intermittency observed in continuous tracking movements. Using spatiotemporal perturbations in humans, we demonstrate that apparently discrete submovements made 2–3 times per second reflect constructive interference between motor errors and continuous feedback corrections that are filtered by intrinsic circuitry in the motor system. Local field potentials in monkey motor cortex revealed characteristic signatures of a Kalman filter, giving rise to both low-frequency cortical cycles during movement, and delta oscillations during sleep. We interpret these results within the framework of optimal feedback control, and suggest that the intrinsic rhythmicity of motor cortical networks reflects an internal model of external dynamics, which is used for state estimation during feedback-guided movement.

**Editorial note:** This article has been through an editorial process in which the authors decide how to respond to the issues raised during peer review. The Reviewing Editor's assessment is that all the issues have been addressed (see decision letter).

DOI: https://doi.org/10.7554/eLife.40145.001

**\*For correspondence:**
andrew.jackson@ncl.ac.uk

**Present address:** [†]Faculty of Medicine, Universitas Indonesia, Jakarta, Indonesia; [‡]Department of Basic Neuroscience, Faculty of Medicine, University of Geneva, Genève, Switzerland

**Competing interests:** The authors declare that no competing interests exist.

## Introduction

Many visually-guided movements are characterized by intermittent speed fluctuations. For example, during the tracking of slowly-moving target, humans make around 2–3 submovements per second. Although first described over a century ago (*Woodworth, 1899*; *Craik, 1947*; *Vince, 1948*) the cause of movement intermittency remains debated. Submovements often disappear in the absence of vision (*Miall et al., 1993a*) and are influenced by feedback delays (*Miall, 1996*), suggesting that their timing depends on extrinsic properties of visuomotor feedback loops. However, some rhythmicity persists in the absence of feedback (*Doeringer and Hogan, 1998*), and it has been suggested that an internal refractory period, clock or oscillator parses complex movements into discrete, isochronal segments (*Viviani and Flash, 1995*; *Loram et al., 2006*; *Hogan and Sternad, 2012*; *Russell and Sternad, 2001*). Cyclical dynamics within motor cortical networks with a time period of 300–500 ms may reflect the neural correlates of such an intrinsic oscillator (*Churchland et al., 2012*; *Hall et al., 2014*). During continuous tracking, each submovement is phase-locked to a single cortical cycle, giving rise to low-frequency coherence between cortical oscillations and movement speed (*Jerbi et al., 2007*; *Hall et al., 2014*; *Pereira et al., 2017*).

It has been proposed that the intrinsic dynamics of recurrently-connected cortical networks act as an 'engine of movement', responsible for internal generation and timing of the descending motor command (*Churchland et al., 2012*). However, another possibility is that low-frequency dynamics observed in motor cortex arise from sensorimotor feedback loops through the external environment. On the one hand, cortical cycles appear conserved across a wide range of behaviors and even share

a common structure with delta oscillations during sleep (*Hall et al., 2014*), consistent with a purely intrinsic origin. On the other hand, the influence of feedback delays on submovement timing suggests an extrinsic contribution to movement intermittency. Therefore, we examined the effect of delay perturbations during visuomotor tracking in humans and monkeys, to dissociate both delay-independent (intrinsic) and delay-dependent (extrinsic) components of movement kinematics and cortical dynamics.

We interpret our findings using stochastic optimal control theory, which has emerged as an influential approach to understanding movement (*Todorov and Jordan, 2002*; *Scott, 2004*). Given noisy, delayed sensory measurements, an optimal feedback controller (OFC) continually estimates the current motor state using an internal model of external dynamics. We show that this can provide a computational framework for understanding both extrinsic and intrinsic contributions to intermittency, accounting for many puzzling features of submovements and providing a parsimonious explanation for conserved cyclical dynamics in motor cortex networks during behavior and sleep.

## Results

### Overview

Our results are organized as follows. First, we describe behavioral results with human subjects, examining the effects of delay perturbations on movement intermittency and feedback responses during an isometric visuomotor tracking task. Second, we introduce a simple computational model to illustrate how principles of optimal feedback control, and in particular state estimation, can explain the key features of our data. Finally, we examine local field potentials recorded from the motor cortex of monkeys performing a similar task, to show that cyclical neural trajectories are consistent with the implementation of state estimation circuitry.

### Submovement frequencies are affected by feedback delays

Our first experiment aimed to characterize the dependence of submovement frequencies on feedback delays. Human subjects generated bimanual, isometric, index finger forces to track targets that moved in slow 2D circular trajectories with constant speed (*Figure 1A*). We measured intermittency in the angular velocity of the cursor (*Figure 1B,C*) using spectral analysis over a 10 s window beginning 5 s after the trial start. Under unperturbed feedback conditions, power spectra generally exhibited a principal peak at around 2 Hz (*Figure 1D*). This frequency was only slightly affected by target speed (*Figure 1—figure supplement 1*), consistent with previous reports (*Miall, 1996*) and perhaps suggestive of an intrinsic oscillator or clock determining submovement timing.

However, submovement frequencies were markedly altered when visual feedback of the cursor was delayed relative to finger forces. With delays of 100 and 200 ms, the frequency of the primary peak reduced to around 1.4 and 1 Hz respectively (*Figure 1D*, *Figure 1—figure supplements 1* and *2*), demonstrating that submovement timing in fact depended on extrinsic feedback properties. Interestingly, a further peak appeared at approximately three times the frequency of the primary peak, and with increased delays, of 300 and 400 ms, a fifth harmonic was observed. The time-periods of the first, third and fifth harmonics were linearly related to extrinsic delay times, with gradients of $1.89 \pm 0.20$, $0.59 \pm 0.04$ and $0.33 \pm 0.11$ respectively (*Figure 1E*, *Table 1*).

These results are consistent with a feedback controller responding to broad-spectrum (stochastic) tracking errors introduced by noise in the motor output, for which the response is delayed by time, $\tau$. In signal-processing terms, subtracting a delayed version from the original signal is known as comb filtering (*Figure 1E*). Although comb filters subtracting in either feedforward or feedback directions have qualitatively similar behavior, we illustrate only the feedforward architecture in *Figure 1E*, as we will later show this to match better the experimental data. For motor noise components with a time period, $T = \frac{\tau}{1}, \frac{\tau}{2}, \frac{\tau}{3} \ldots$, delayed corrections accurately reflect current errors, resulting in regularly spaced notches in the amplitude response of the system (*Figure 1G*) and attenuation in the resultant cursor movement through destructive interference. By contrast, for motor noise with a time-period, $T = \frac{2\tau}{1}, \frac{2\tau}{3}, \frac{2\tau}{5} \ldots$, delayed corrections are exactly out-of-phase with the current error. Thus, corrective movements exacerbate these components through constructive interference, leading to spectral peaks at frequencies:

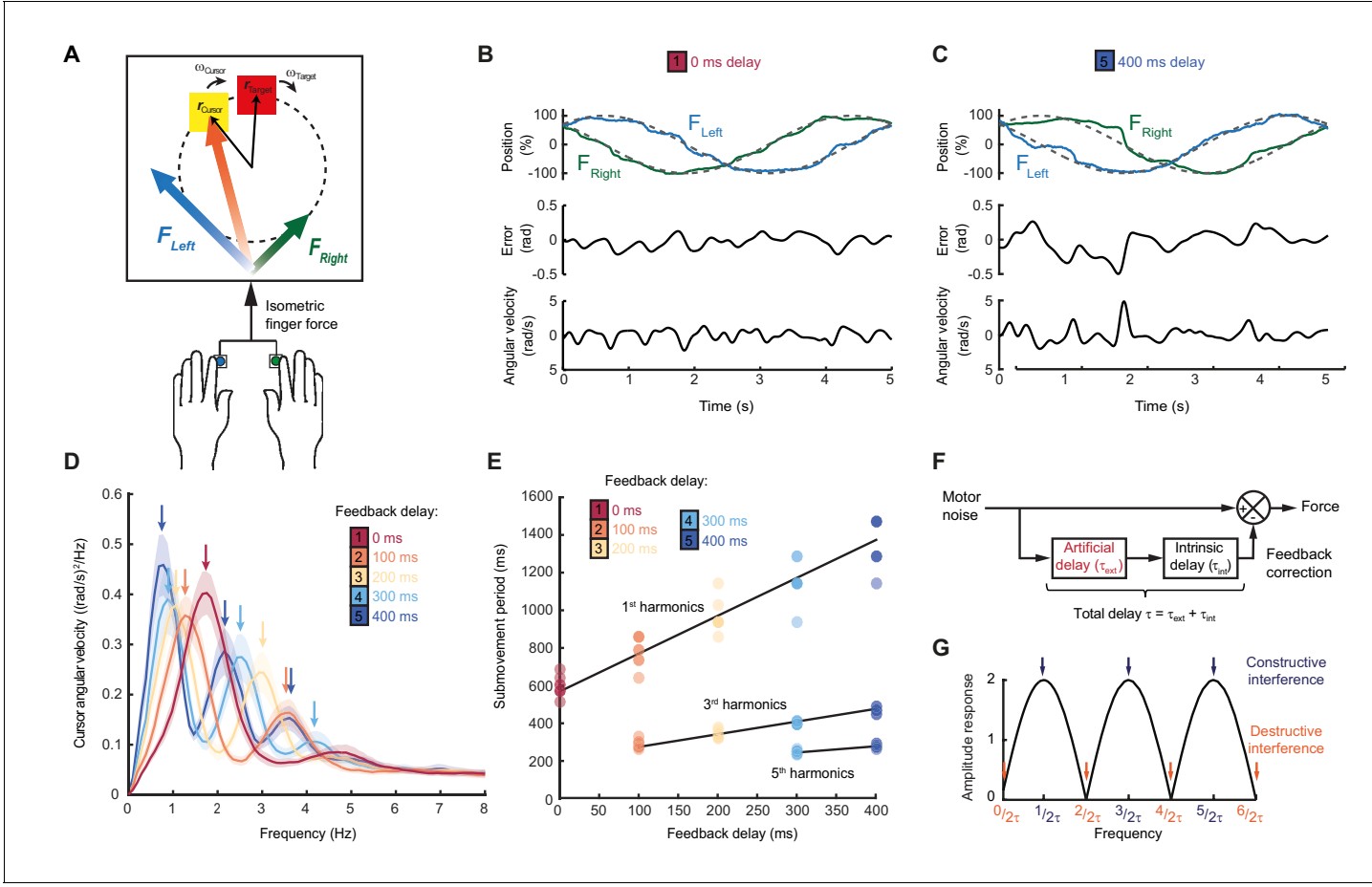

**Figure 1.** Movement intermittency during visuomotor tracking depends on feedback delays. (A) Schematic of human tracking task. Bimanual isometric finger forces controlled 2D cursor position to track slow, circular target motion. Kinematic analyses used the angular velocity of the cursor subtended at the screen center during the middle of each trial. (B) Example force (*top*), angular error (*middle*) and cursor angular velocity (*bottom*) traces during target tracking with no feedback delay. Submovements are evident as intermittent fluctuations in angular velocity. (C) Example movement traces with 400 ms feedback delay. (D) Power spectra of cursor angular velocity with different feedback delays between 0 and 400 ms. Analysis based on a 10 s window beginning 5 s after trial start. Average of 8 subjects, shading indicates standard error of mean (s.e.m.). See also *Figure 1—figure supplement 2*. (E) Submovement periods (reciprocal of the peak frequency for each harmonic) for all subjects with different feedback delays. Lines indicate linear regression over all subjects. See *Table 1* for summary of individual subject regression analysis. (F) Schematic of a simple delayed feedback controller. (G) Amplitude response of the system shown in (F), known as a comb filter.

DOI: https://doi.org/10.7554/eLife.40145.002

The following source data and figure supplements are available for figure 1:

**Source data 1.** Subject information, time periods of submovement peaks and associated regression analysis.

DOI: https://doi.org/10.7554/eLife.40145.006

**Figure supplement 1.** Effect of target speed on movement intermittency.

DOI: https://doi.org/10.7554/eLife.40145.003

**Figure supplement 2.** Individual subject power spectra of cursor velocity with different feedback delays.

DOI: https://doi.org/10.7554/eLife.40145.004

**Figure supplement 3.** Trajectory variability depends on change in isometric force.

DOI: https://doi.org/10.7554/eLife.40145.005

$$f = \frac{1}{T} = \frac{N}{2(\tau_{\mathrm{int}} + \tau_{\mathrm{ext}})} \quad \text{With } N = 1, 3, 5 \ldots \tag{1}$$

Submovement frequencies in our data approximately matched this model, assuming the total feedback delay comprised the experimental manipulation $\tau_{\mathrm{ext}}$ added to a constant physiological response latency $\tau_{\mathrm{int}}$ of around 300 ms (*Table 1*), comparable to visual reaction times.

**Table 1.** The dependency of submovement period on feedback delay.

Shown in the table are the gradients and intercepts of regression lines fitted to each harmonic group in *Figure 1E*. The time period of each spectral peak was regressed against feedback delay. Shown in square brackets are 95% confidence intervals of these values. Also shown is the estimated intrinsic time delay calculated using *Equation (1)*.

| Harmonic (*N*) | Predicted slope = 2/*N* | Measured slope | Measured intercept (ms) | $R^2$ | *P* | $\tau_{int}$ = Intercept*N/2 |
|---|---|---|---|---|---|---|
| 1 | 2 | 1.89 [1.69,2.09] | 589 ms [539,638] | 0.90 | <0.00001 | 294 ms [270,319] |
| 3 | 0.67 | 0.59 [0.53,0.65] | 226 ms [211,242] | 0.94 | <0.00001 | 340 ms [316,362] |
| 5 | 0.4 | 0.33 [0.22,0.45] | 146 ms [106,185] | 0.75 | <0.00001 | 364 ms [266,463] |

DOI: https://doi.org/10.7554/eLife.40145.007

## Submovements occur at frequencies of constructive interference between motor errors and delayed corrections

According to this extrinsic interpretation, intermittency arises not from active internal generation of discrete submovement events, but as a byproduct of continuous, linear feedback control with inherent time delays. Submovement frequencies need not be present in the smooth target movement, nor do they arise from controller non-linearities. Instead these frequencies reflect components of broad-band motor noise that are exacerbated by constructive interference with delayed feedback corrections. To seek further evidence that intermittency arises from such constructive interference, we performed a second experiment in which artificial errors were generated by spatial perturbation of the cursor. Within individual trials, a sinusoidal displacement was added to the cursor position in a direction aligned to target motion and at a frequency between 1 and 5 Hz. Perturbation amplitudes were scaled to have equivalent peak angular velocities (equal to the angular velocity of the target). Our hypothesis was that artificial errors at submovement frequencies would be harder to track (because of constructive interference) than perturbations at frequencies absent from the velocity spectrum.

*Figure 2A* shows example tracking behavior with a 2 Hz perturbation. Note that the peak angular velocity of force responses (*black line*, calculated from the subject's finger forces) occurred around the same time as the peak angular velocity of the perturbation (*green line*). As a result, the angular velocity of the cursor (*yellow line*, reflecting the combination of the subject's forces with the perturbation) exhibited pronounced oscillations that were larger than the perturbation. *Figure 2B* shows performance in the same task when visual feedback was delayed by 200 ms. In this condition, peaks in force velocity coincided with perturbation troughs, attenuating the disturbance to cursor velocity. *Figure 2C,D* and *Figure 2—figure supplement 1* overlay cursor velocity spectra in the presence of each perturbation frequency (with feedback delays of 0 and 200 ms), again calculated over a 10 s window beginning 5 s after the trial start. As previously, in the absence of feedback delay, the frequency of submovements was around 2 Hz. Correspondingly, perturbations at 2 Hz induced a large peak in the cursor velocity spectrum, indicating that the artificial error was not effectively tracked. By contrast, with a feedback delay of 200 ms the cursor velocity spectrum with a 2 Hz perturbation was attenuated. The largest spectral peaks were instead associated with 1 and 3 Hz perturbations, matching the frequencies of submovements in this delay condition.

*Figure 2E* shows the amplitude response of cursor movements at each frequency for both delay conditions. Unlike a power spectrum, the cursor amplitude response measures only cursor movements that are phase-locked to the perturbation (normalized by the perturbation amplitude), and therefore estimates the overall transfer function of the closed-loop control system. Cursor amplitude responses greater than unity at 2 Hz (with no delay), and at 1 and 3 Hz (with 200 ms delay) indicate exacerbation of intermittencies introduced by artificial errors at submovement frequencies. Analysis of variance (ANOVA) with two factors (delay time and perturbation frequency) revealed a highly significant interaction (n = 8 subjects, $F_{4,70}$=110.2, p<0.0001), confirming the interdependence of feedback delays and frequencies of constructive/destructive interference.

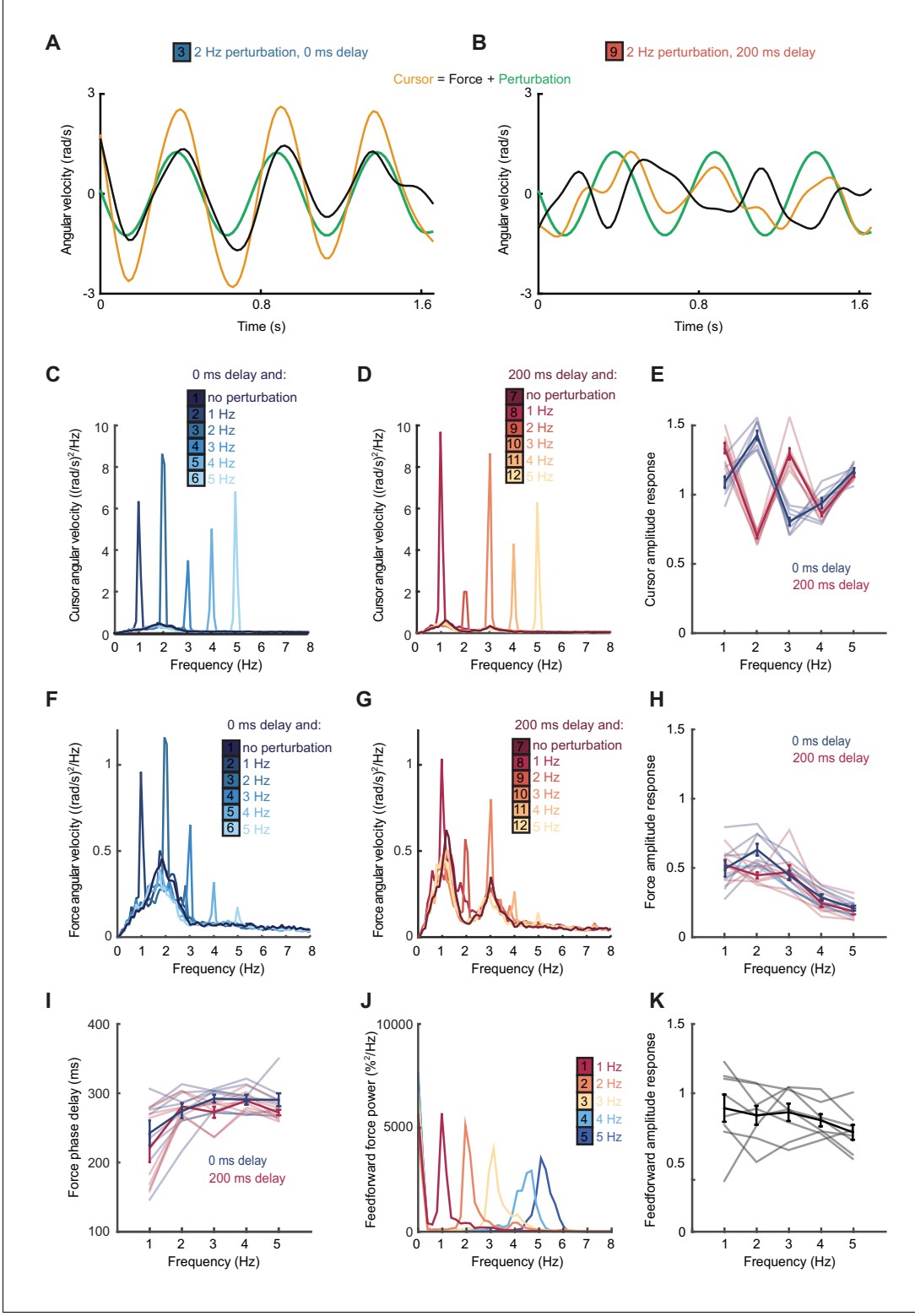

**Figure 2.** Frequency responses and phase delays to artificial motor errors. (**A**) Example force (*black*) and cursor (*yellow*) angular velocity traces in the presence of a 2 Hz perturbation (*green*) when no feedback delay was added. The force response and perturbation sum to produce large fluctuations in cursor velocity. (**B**) Comparable data with a feedback delay of 200 ms. In this condition, force responses cancel the perturbation leading to an attenuation of intermittency. (**C**) Power spectra of cursor angular velocity with 1–5 Hz perturbations and no feedback delay. Analysis based on 10 s windows beginning 5 s after trial start. Average of 8 subjects. See also

*Figure 2 continued on next page*

*Figure 2 continued*

***Figure 2—figure supplement 1***. (D) Power spectra of cursor angular velocity with 1–5 Hz perturbations and 200 ms feedback delay. (E) Cursor amplitude response to 1–5 Hz perturbations with no feedback delay (*blue*) and 200 ms feedback delay (*red*) for individual subjects. Also shown is the average ± s.e.m. of 8 subjects. (F) Power spectra of force angular velocity with 1–5 Hz perturbations and no feedback delay. See also ***Figure 2—figure supplement 2***. (G) Power spectra of force angular velocity with 1–5 Hz perturbations and 200 ms feedback delay. (H) Force amplitude response to 1–5 Hz perturbations with no feedback delay (*blue*) and 200 ms feedback delay (*red*). Also shown is average ± s.e.m. of 8 subjects. (I) Intrinsic phase delay of force response to 1–5 Hz perturbations with no feedback delay (*blue*) and 200 ms feedback delay (*red*). Also shown is average ± s.e.m. of 8 subjects. (J) Power spectrum of finger forces generated in the feedforward task with auditory cues at 15 Hz. Average of 8 subjects. See also ***Figure 2—figure supplement 3***. (K) Force amplitude response to auditory cues in the feedforward task. Also shown is average ± s.e.m. of 8 subjects. All analyses based on a 10 s windows beginning 5 s after trial start.
DOI: https://doi.org/10.7554/eLife.40145.008

The following source data and figure supplements are available for figure 2:

**Source data 1.** Subject information, perturbation responses and feedforward amplitude responses.
DOI: https://doi.org/10.7554/eLife.40145.012

**Figure supplement 1.** Individual subject power spectra of cursor velocity with perturbations.
DOI: https://doi.org/10.7554/eLife.40145.009

**Figure supplement 2.** Individual subject power spectra of force velocity with perturbations.
DOI: https://doi.org/10.7554/eLife.40145.010

**Figure supplement 3.** Feedforward task.
DOI: https://doi.org/10.7554/eLife.40145.011

## Intrinsic dynamics in the visuomotor feedback loop

Although submovement frequencies depended on extrinsic feedback delays, examination of the velocity spectra in *Figure 1D* suggests that intermittency peaks were embedded within a broad, delay-independent low-pass envelope. This envelope could simply reflect the power spectrum of tracking errors (i.e. motor noise is dominated by low-frequency components). However, an additional possibility is that the gain of the feedback controller varies across frequencies (e.g. low-frequency noise components generate larger feedback corrections). To explore the latter directly, we examined subjects' force responses to our artificial perturbations.

*Figure 2F,G* and *Figure 2—figure supplement 2* show power spectra of the angular velocity derived from subjects' forces, under feedback delays of 0 and 200 ms. Note that this analysis differs from *Figure 2C,D* in that we now consider only the forces generated by the subjects, rather than the resultant cursor movement (which combines these forces with the perturbation). *Figure 2H* shows the corresponding force amplitude response for each perturbation frequency. The force amplitude response measures only force responses that are phase-locked to the perturbation (normalized by the perturbation amplitude) and is related to the transfer function within the feedback loop. Unlike the cursor amplitude responses described previously, force amplitude responses were largely independent of extrinsic delay. However, as with the velocity spectra in *Figure 1D*, feedback gains were also attenuated at higher frequencies. A two-factor ANOVA confirmed a significant main effect of frequency (n = 8 subjects, $F_{4,70}$=36.3, p<0.0001) but not delay time ($F_{1,70}$=3.1, p=0.08), and only a weakly significant interaction ($F_{4,70}$=2.9, p=0.03). In other words, feedback corrections to artificial errors revealed a delay-independent filter matching the attenuation of submovement peaks at higher frequencies.

Interestingly, the phase delay of force responses was also influenced by perturbation frequency (*Figure 2I*). Effectively, corrections to low-frequency perturbations occurred slightly earlier than those to higher frequencies, indicating a predictive component to feedback responses. As with the amplitude response, there was a significant effect of frequency ($F_{4,70}$=9.5, p<0.0001) but not extrinsic delay ($F_{1,70}$ =2.6, p=0.12) on this phase delay, and no significant interaction ($F_{4,70}$=0.7, p=0.6).

We next considered whether high-frequency attenuation of feedback responses was a property of motor pathways, for example reflecting filtering by the musculoskeletal system. However, it is well-known that the frequencies of feedforward movements can readily exceed submovement frequencies observed during feedback-guided behavior (*Kunesch et al., 1989*). We confirmed this by asking subjects to produce force fluctuations of a defined amplitude, but without providing a

moving target to track. Instead we used auditory cues (a metronome) to indicate the required movement frequency. In this case, subjects could generate force fluctuations up to 5 Hz with little attenuation (*Figure 2J,K* and *Figure 2—figure supplement 3*). Therefore we concluded that filtering of corrective responses was not an inherent property of feedforward motor pathways but instead reflected intrinsic dynamics in the visuomotor feedback loop.

## Intrinsic dynamics and optimal state estimation

The visual system can perceive relatively high frequencies (up to flicker-fusion frequencies above 10 Hz). However, while feedforward movements can in some cases approach these frequencies, discrepancies while tracking slowly-moving objects in the physical world are unlikely to change quickly. The limbs and real-world targets will tend (to a first approximation) to move with a constant velocity unless acted upon by a force. Moreover, even for isometric tasks, the drift in force production is dominated by low-frequency components (*Baweja et al., 2009*; *Slifkin and Newell, 2000*), possibly consistent with neural integration in the descending motor pathway (*Shadmehr, 2017*). Given inherent uncertainties in sensation, an optimal state estimator should attribute high-frequency errors to sensory noise (which is unconstrained by Newtonian and/or neuromuscular dynamics).

Formally, the task of distinguishing the true state of the world from uncertain, delayed measurements can be achieved by a Kalman filter, which continuously integrates new evidence with updated estimates of the current state, evolving according to a model of the external dynamics (*Figure 3A*). For simplicity we used Newtonian dynamics, although similar results would likely be obtained for other second-order state transition models. We assumed the 1D position of the body (cursor) relative to the target should evolve with constant velocity unless acted upon by accelerative forces, leading to the state transition model:

$$\begin{bmatrix} x_k \\ v_k \end{bmatrix} = \begin{bmatrix} 1 & \Delta t \\ 0 & 1 \end{bmatrix} \begin{bmatrix} x_{k-1} \\ v_{k-1} \end{bmatrix} + \begin{bmatrix} 0 \\ \Delta t \end{bmatrix} a_k \tag{2}$$

where $x_k$ and $v_k$ are the relative position and velocity of the cursor at time-step $k$, $\Delta t$ is the interval between time-steps, and the process noise $a_k \sim N(0, \sigma_a^2)$. Visual feedback, $y_k$, was assumed to comprise a noisy measurement of relative position:

$$y_k = x_k + \varepsilon_k \tag{3}$$

with measurement noise $\varepsilon_k \sim N(0, \sigma_\varepsilon^2)$.

Optimal estimates of relative position and velocity, $\widehat{x}_k$ and $\widehat{v}_k$ are given by a steady-state Kalman filter of the form:

$$\begin{bmatrix} \widehat{x}_k \\ \widehat{v}_k \end{bmatrix} = \begin{bmatrix} 1 - K_{\mathrm{pos}} & \Delta t \\ -K_{\mathrm{vel}} & 1 \end{bmatrix} \begin{bmatrix} \widehat{x}_{k-1} \\ \widehat{v}_{k-1} \end{bmatrix} + \begin{bmatrix} K_{\mathrm{pos}} \\ K_{\mathrm{vel}} \end{bmatrix} y_{k-1} \tag{4}$$

The innovation gains $K_{\mathrm{pos}}$ and $K_{\mathrm{vel}}$ depend only on the ratio of (accelerative) process to (position) measurement noise, $\rho = \frac{\sigma_a}{\sigma_\varepsilon}$, which in turn determines the cut-off frequency above which measurements are filtered $(\sim \frac{1}{2\pi} \sqrt{\rho})$. *Figure 3B,C* shows the amplitude response for position and velocity estimates produced by the Kalman filter. Since these are out of phase with each other, their cross-spectral density (which captures the amplitude and phase-difference between frequency components common to both signals) will generally be complex. Broadband input therefore results in an imaginary component to this cross-spectrum with a characteristic low-frequency resonance peak determined by the state estimator dynamics (*Figure 3D*).

Feedback delays can be accommodated by projecting the state estimate forward in time:

$$\widehat{z}_k = \begin{bmatrix} 1 & \tau_{\mathrm{int}} \end{bmatrix} \begin{bmatrix} \widehat{x}_k \\ \widehat{v}_k \end{bmatrix} \tag{5}$$

The phase delay of the optimal position estimate of the current state, $\widehat{z}_k$, falls towards zero at low frequencies, consistent with a predictive component when interpreting low-frequency errors (*Figure 3E*).

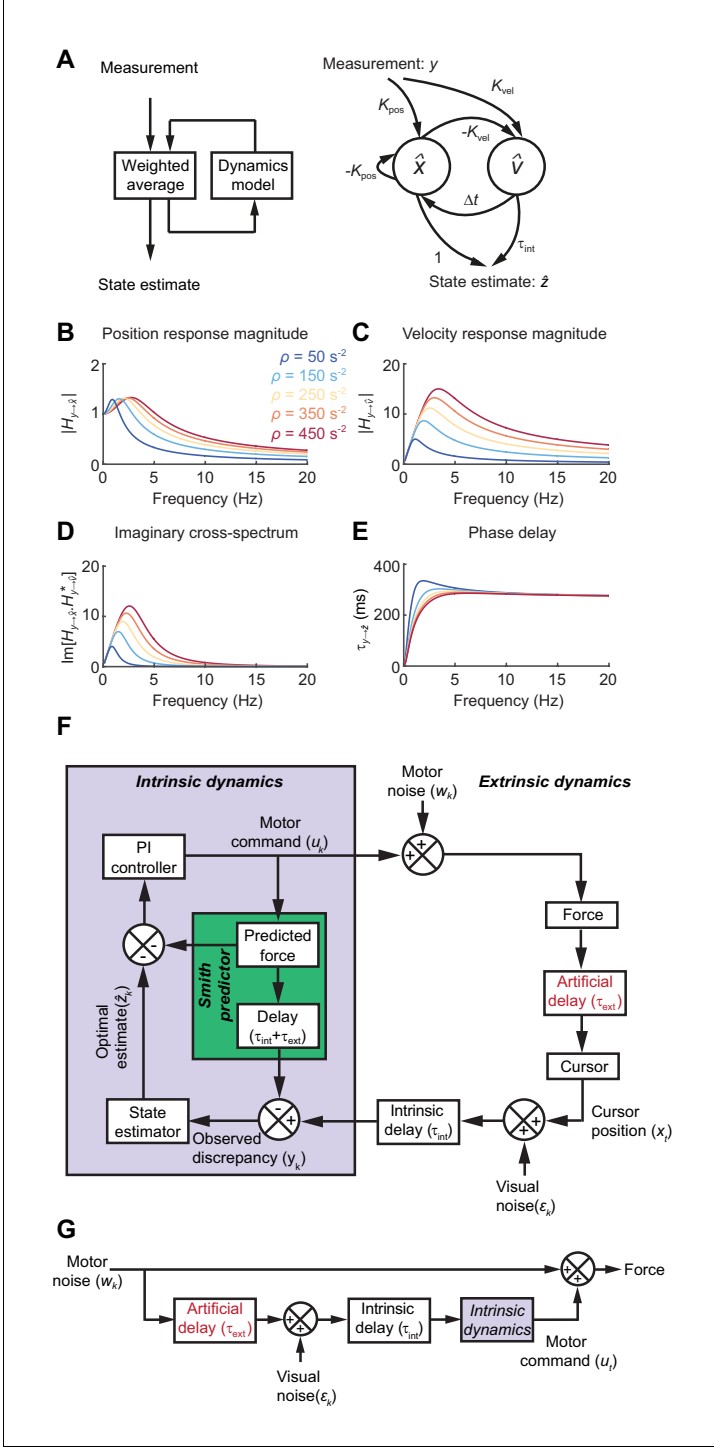

**Figure 3.** State estimation with a Kalman filter. (**A**) *Left:* Schematic of a Kalman filter. Noisy measurements are combined with an internal model of the external dynamics to update an optimal estimate of current state. *Right:* A dynamical system for optimal estimation of position, based on an internal model of position and velocity. (**B, C**) Magnitude response of transfer function from measurement to position and velocity estimates, respectively, for a Kalman filter with different ratios of process to measurement noise ($\rho$). (**D**) Imaginary component of cross-spectrum between position and velocity transfer functions. (**E**) Phase delay of optimal estimate of position based on delayed measurement of position. (**F**) Schematic of optimal feedback controller model incorporating state estimation and a Smith Predictor architecture to accommodate feedback delays. (**G**) Simplified rearrangement of (**F**), showing the

*Figure 3 continued*

feedforward relationship between motor noise and force output. This rearrangement is possible because the Smith Predictor prevents motor corrections reverberating multiple times around the feedback loop.

DOI: https://doi.org/10.7554/eLife.40145.013

## Incorporating intrinsic and extrinsic dynamics in a model of movement intermittency

To illustrate how such a steady-state Kalman filter can account for the main features of our human behavioral data, we incorporated it within a simple 1D feedback controller (*Figure 3F*; see Materials and methods for details). We included an internal feedback loop to cancel the sensory consequences of motor commands, known as a Smith Predictor (*Abe and Yamanaka, 2003*; *Miall et al., 1993b*). This prevents corrections from reverberating around the external feedback loop, such that the resultant closed-loop behavior is formally equivalent to the simpler feedforward comb filter shown in *Figure 3G*. This rearrangement provides a useful intuition about our behavioral results. Tracking errors (due to motor noise) drive feedback corrections that are delayed, corrupted (by sensory noise) and filtered (by intrinsic dynamics). The power spectrum of the resultant movements reflects constructive/destructive interference between feedback corrections and the original tracking error.

The Smith Predictor model readily accounted for the main features of our human data, including the cursor amplitude response to perturbations (*Figure 4A–E*), and the low-pass filtering (*Figure 4F–H*) and phase delay (*Figure 4I*) of force responses. Moreover, because of frequency-dependent phase delays introduced by state estimation, the model suggested that precise frequencies of submovement peaks should deviate slightly from those calculated using a constant physiological response latency. Effectively, a predictive state estimator responding more quickly to low-frequency errors behaves like a feedback controller with a reduced loop delay. This effect was confirmed in our behavioral data by calculating (with *Equation (1)*) the intrinsic delay time corresponding to each spectral peak under all feedback delay conditions in our first experiment (arrows in *Figure 1D*). Rather than being a constant, this intrinsic delay time was positively correlated with frequency (n = 11 spectral peaks, R = 0.78, p=0.0046) and matched well the frequency-dependent phase delay predicted by our model (*Figure 4J*).

Finally, overall tracking performance (as measured by the root mean squared positional error over time) matched well with subjects' actual performance across conditions in our second experiment (*Figure 4K*). Note that, irrespective of delay, the lowest frequency perturbation was associated with the greatest positional error, since perturbations had equal peak-to-peak velocity and were therefore larger in amplitude at low frequencies. However, performance was most affected by the 1 Hz perturbation with a 200 ms delay, corresponding to a frequency of constructive interference.

## Emergence of delay-specific predictive control during individual trials

While not simulating the full complexity of upper-limb control, our model was intended to illustrate the interplay between intrinsic and extrinsic dynamics during tracking. More sophisticated models would likely exhibit qualitatively similar behavior, so long as they also incorporated extrinsic, delay-dependent feedback and intrinsic, delay-independent dynamics. However, in order to generate stable tracking behavior, the Smith Predictor architecture requires accurate compensation for external delays within an internal feedback loop. For this to be a plausible model to explain our data, the controller would need to adapt quickly to new extrinsic delay conditions that varied from trial to trial in our experiment. Therefore we were interested in whether we could observe such rapid adaptation of control strategies within individual trials. Moreover, we asked whether this adaptation was delay-specific as expected for a Smith Predictor, or could perhaps be explained by a simpler delay-independent feedback controller.

*Figure 5A,B* compares schematics of two linear feedback controllers with and without the internal Smith Predictor loop. Both incorporate intrinsic dynamics, and as a result of extrinsic feedback delays exacerbate motor noise at frequencies given by *Equation (1)*. However, the comb filter rearrangements in *Figure 5C,D* show how the two architectures predict different relationships between the feedback gain resulting from this intrinsic dynamics, $G(i\omega)$, and the closed-loop force amplitude

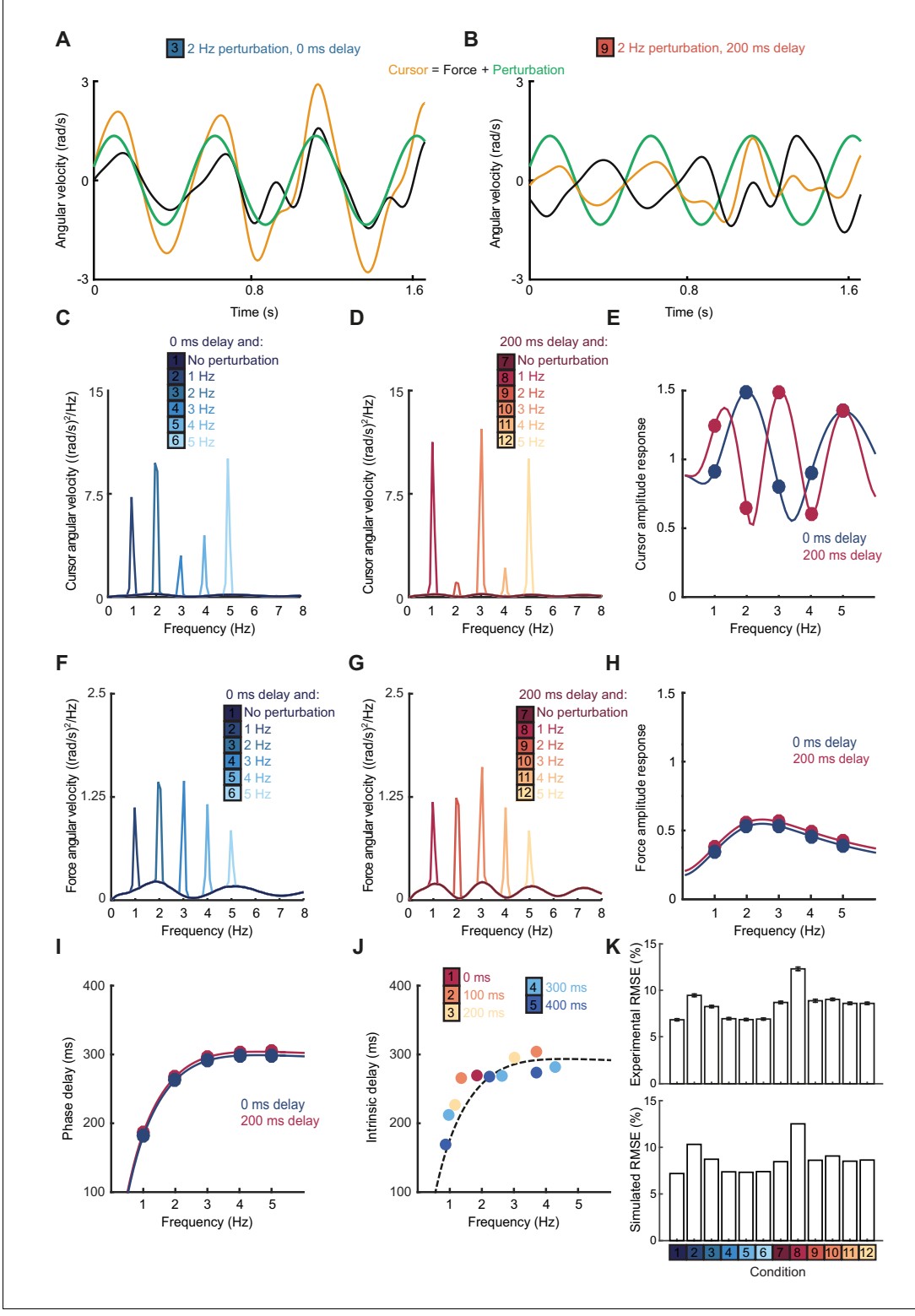

**Figure 4.** Smith Predictor model with optimal state estimation reproduces human behavioral data. (**A**) Simulated tracking performance of the model with a 2 Hz sinusoidal perturbation and no feedback delay. (**B**) Simulated tracking performance of the model with a 2 Hz sinusoidal perturbation and 200 ms feedback delay. (**C**) Power spectrum of simulated cursor velocity with 1–5 Hz perturbations and no feedback delay. (**D**) Power spectrum of simulated cursor velocity with 1–5 Hz perturbations and 200 ms feedback delay. (**E**) Simulated cursor amplitude response to 1–5 Hz perturbations with no feedback delay (*blue*) and 200 ms feedback delay (*red*). (**F**) Power

*Figure 4 continued on next page*

*Figure 4 continued*

spectrum of simulated force velocity with 1–5 Hz perturbations and no feedback delay. (G) Power spectrum of simulated force velocity with 1–5 Hz perturbations and 200 ms feedback delay. (H) Simulated force amplitude response to 1–5 Hz perturbations with no feedback delay (*blue*) and 200 ms feedback delay (*red*). (I) Simulated intrinsic phase delay of force responses to 1–5 Hz perturbations with no feedback delay (*blue*) and 200 ms feedback delay (*red*). (J) Intrinsic delay times corresponding to all submovement peaks/harmonics in *Figure 1D*, plotted against the frequency of the peak. Dashed line indicates phase delay of the simulated optimal controller (K) *Top:* Positional inaccuracy of human tracking for all conditions quantified as root mean squared error (RMSE). Average ± s.e.m. of 8 subjects. *Bottom:* RMSE of simulated tracking for all conditions.
DOI: https://doi.org/10.7554/eLife.40145.014

response to perturbations, $H_{force}(i\omega)$. We asked how each model explained the experimental data by inferring the intrinsic dynamics that would be required to generate our observed force amplitude responses under both architectures.

*Figure 5E* shows the intrinsic gain that a simple feedback controller would need to explain the amplitude responses observed during 5 s sections of experimental data taken from the start, middle and end of each trial. While the general pattern was one of low-pass filtering, the intrinsic dynamics inferred for each delay condition diverged progressively through the trial. Therefore we can conclude that the control strategy used by subjects was indeed adapting during a single trial, and that this adaptation was delay-specific. Interestingly, intrinsic feedback gains inferred using the Smith Predictor model (*Figure 5F*) became progressively more similar as the trial progressed. Therefore the adaptation process could parsimoniously be interpreted as the emergence of an appropriately-calibrated Smith Predictor with delay-independent intrinsic dynamics, as predicted by our optimal control model. The time-course of this adaptation (*Figure 5G*) was associated with a reduction in both low-frequency phase delays (*Figure 5H*) and the average lag of the cursor behind the target (*Figure 5I*), showing that subjects quickly learned to compensate for feedback delays within individual trials.

## Movement intermittency in a non-human primate tracking task

The amplitude and phase responses to perturbations during human visuomotor tracking provided evidence for intrinsic low-frequency dynamics in feedback corrections, which we have interpreted in the framework of optimal state estimation. The schematic on the right of *Figure 3A* suggests how a simple Kalman filter could be implemented by neural circuitry, with two neural populations (representing position and velocity) evolving according to *Equation (4)* and exhibiting a resonant cross-spectral peak (*Figure 3D*). To seek further evidence for the neural implementation of such a filter we turned to intracortical recordings in non-human primates. We were interested in whether cyclical motor cortex trajectories could reflect the delay-independent dynamics of the two interacting neural populations described above, and thereby account for filtering of feedback responses during visuomotor tracking.

We analyzed local field potential (LFP) recordings from monkey primary motor cortex (M1) during a center-out isometric wrist torque task, which we have used previously to characterize both submovement kinematics and population dynamics (*Hall et al., 2014*). *Figure 6* shows example tracking behavior (*Figure 6A*), radial cursor velocity (*Figure 6B*) and multichannel LFPs (*Figure 6C*) as monkeys moved to peripheral targets under two feedback delay conditions. Movement intermittency was apparent as regular submovement peaks in the radial cursor velocity. Moreover, LFPs exhibited low-frequency oscillations during movement, with a variety of phase-shifts present on different channels. Principal component analysis (PCA) yielded two orthogonal components of the cortical cycle (*Figure 6E*), and the close coupling with submovements was revealed by overlaying the cursor velocity profile onto, in this case, the second principal component (PC) (*Figure 6E*).

## Intrinsic cortical dynamics are unaffected by feedback delays

As with humans, in the absence of feedback delay the cursor velocity (after removing task-locked components, see Materials and methods) was dominated by a single spectral peak (*Figure 7A,E*; *top red traces*). A broad peak at approximately the same frequency was also observed in average LFP power spectra (*Figure 7B,F*). Additionally, we used coherence analysis to confirm consistent

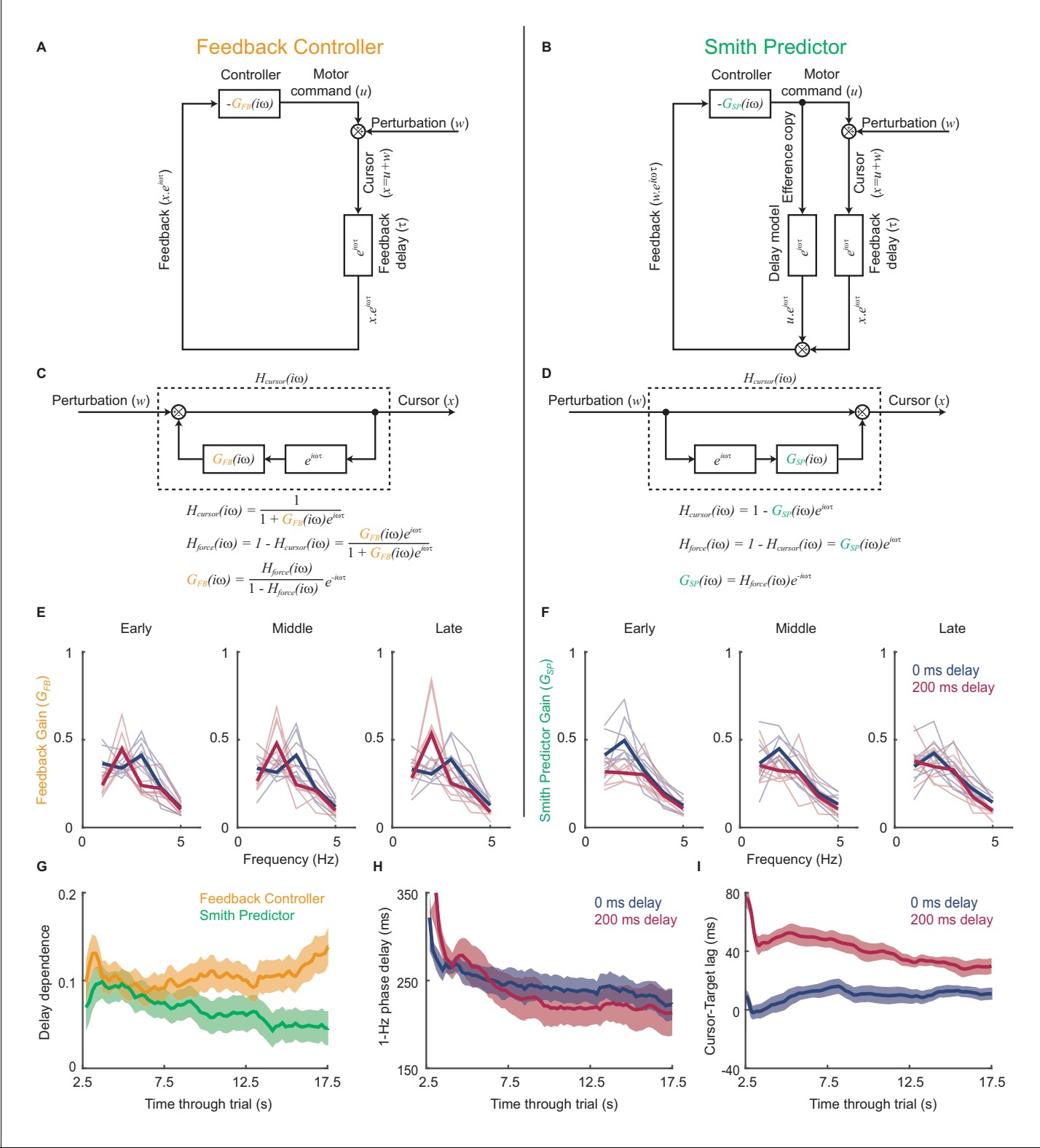

**Figure 5.** Emergence of predictive control strategies within individual trials. (A) Schematic of a simple feedback controller with intrinsic gain, $G_{FB}(i\omega)$ and time delay, τ. (B) A Smith Predictor with intrinsic gain, $G_{SP}(i\omega)$, time delay and calibrated internal feedback loop. (C,D) Rearrangements of the two model architectures to allow derivation of the cursor response function, $H_{cursor}(i\omega)$, and force amplitude response, $H_{force}(i\omega)$. Note that both act as 'comb filters' and exhibit delay-dependent submovement peaks. However the architectures predict different relationships between intrinsic gain and force amplitude response. (E) Feedback gains inferred from experimental data assuming the simple feedback controller architecture, for 5 s windows

*Figure 5 continued on next page*

*Figure 5 continued*
early, middle and late in each trial. Thick line shows average over 8 subjects. Note that feedback gains for different delay conditions become less similar as the trial progresses. (F) Feedback gains inferred from experimental data assuming the Smith Predictor architecture. Feedback gains for different delay conditions become more similar as trial progresses. (G) Delay-dependence of feedback gain (mean-squared difference between delay conditions) inferred from the two architectures. The analysis used a 5 s sliding window through the entire trial. Shading indicates s.e.m. over 8 subjects. (H) Phase delay of feedback gain at 1 Hz inferred from Smith Predictor architecture through trials with 0 and 200 ms delay. (I) Average time lag between cursor and target through trials with 0 and 200 ms delay (and no spatial perturbation).
DOI: https://doi.org/10.7554/eLife.40145.015

phase-coupling between LFPs and cursor velocity (*Figure 7C,G*). Finally, we calculated imaginary coherence spectra between pairs of LFPs (see Materials and methods). The imaginary component of coherence indicates frequencies at which there is consistent out-of-phase coupling in the LFP cross-spectrum. This effectively separates locally-varying oscillatory components from in-phase background signals (e.g. due to volume conduction from distant sources), and revealed more clearly the

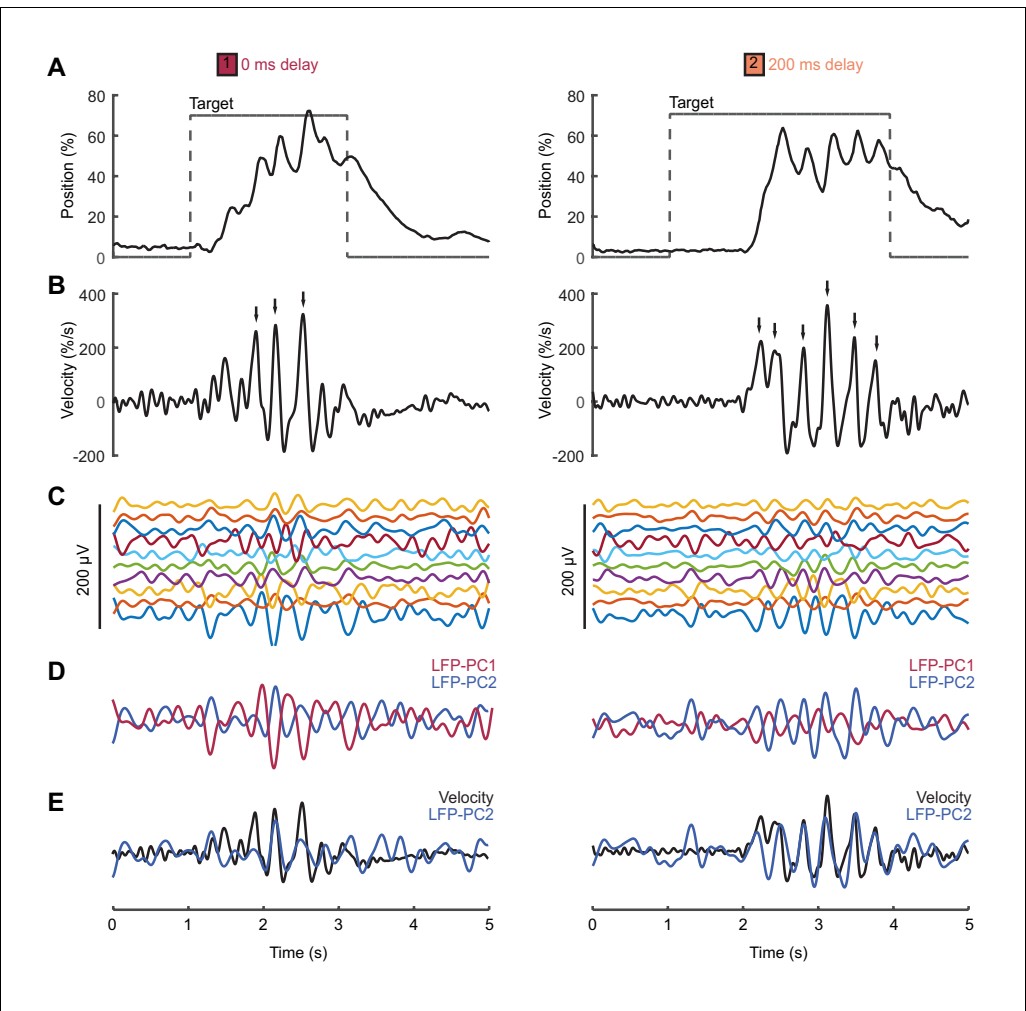

**Figure 6.** Movement intermittency in a non-human primate tracking task. (A) Radial cursor position during a typical trial of the center-out isometric wrist torque task under two different feedback delay conditions. Data from Monkey U. (B) Radial cursor velocity. Arrowheads indicate time of submovements identified as positive peaks in radial cursor velocity. (C) Low-pass filtered, mean-subtracted LFPs from M1. (D) First two principal components (PCs) of the LFP. (E) The second LFP-PC overlaid on the radial cursor velocity.
DOI: https://doi.org/10.7554/eLife.40145.016

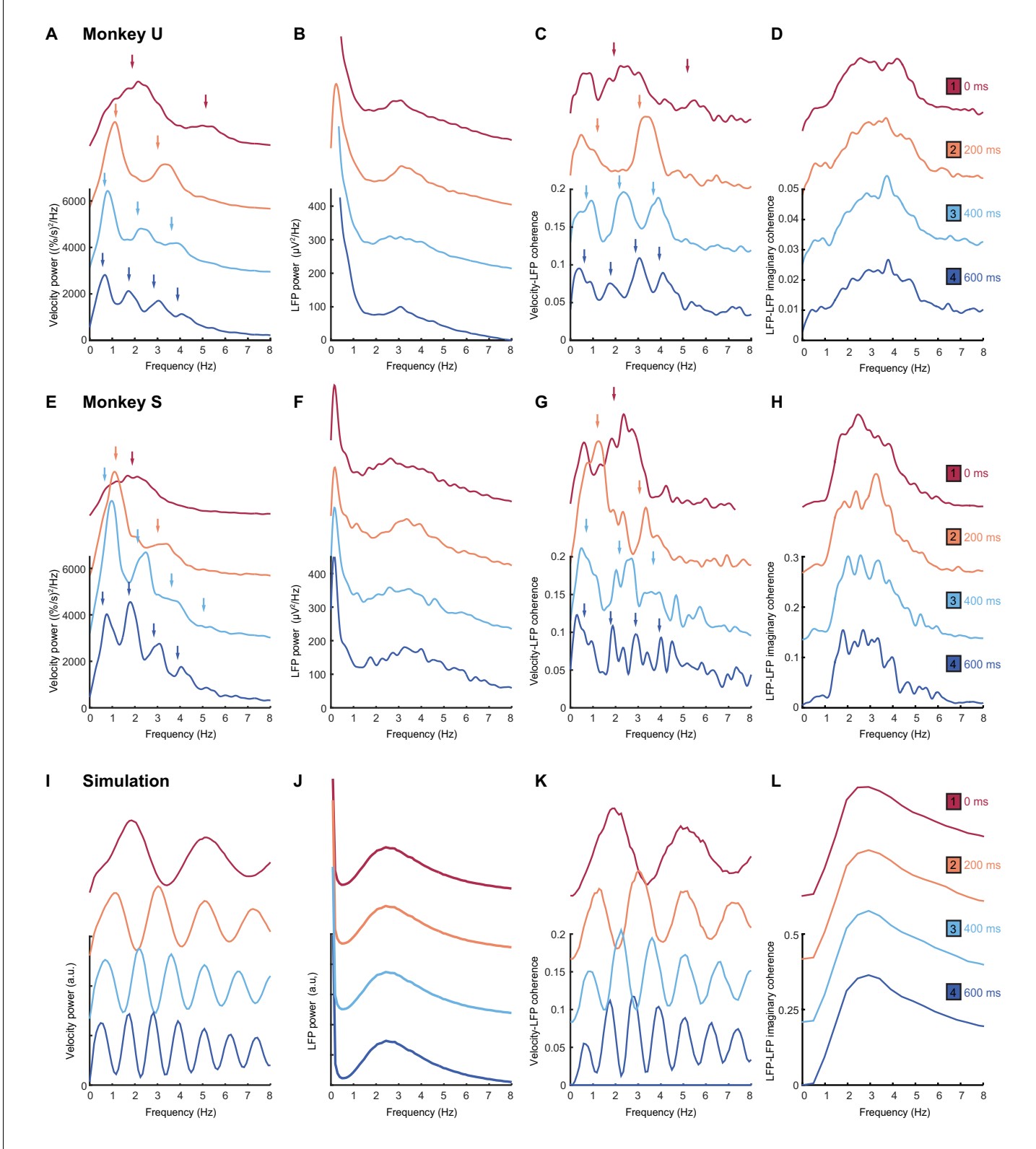

**Figure 7.** Frequency-domain analysis reveals delay-dependent and delay-independent spectral features. (**A**) Power spectrum of radial cursor speed with 0–600 ms feedback delay. Traces have been off-set for clarity. Arrows indicate expected frequencies of peaks from OFC model. Data from Monkey U. (**B**) Average power spectrum of M1 LFPs. (**C**) Average coherence spectrum between radial cursor speed and all M1 LFPs. (**D**) Average imaginary

*Figure 7 continued on next page*

*Figure 7 continued*

coherence spectrum between all pairs of M1 LFPs. (E–H) As above, but for Monkey S. (I–L) Simulated power and coherence spectra produced by the OFC model.

DOI: https://doi.org/10.7554/eLife.40145.017

LFP rhythmicity (*Figure 7D,H*). Note that for no feedback delay, all spectra contain a single peak at around 2–3 Hz.

An obvious interpretation of these results would be that oscillatory activity in the motor system drives submovements in a feedforward manner. In this case, we would expect the frequency of the cortical oscillation to reliably reflect the intermittency observed in behavior. With increasing feedback delays, submovement peaks in monkeys (*Figure 7A,E*; *lower traces*) exhibited a pattern similar to that seen with human subjects. The fundamental frequency was reduced, while odd harmonics grew more pronounced as they came below about 4 Hz. Moreover, coherence spectra between cursor velocity and LFP (*Figure 7C,G*) revealed delay-dependent peaks at both fundamental and harmonic frequencies. Surprisingly however, the power spectrum of the LFP (*Figure 7B,F*) was unaffected by feedback delay, with a single broad peak in the delta band persisting throughout. Moreover, imaginary coherence spectra between pairs of LFPs were also unchanged (*Figure 7D,H*). These results are incompatible with the hypothesis that motor cortical oscillations drive movement intermittency directly, and instead demonstrate a dissociation between delay-dependent submovements and delay-independent cortical dynamics.

We next identified submovements from peaks in the radial cursor speed, in order to examine the temporal profile of their associated LFPs. Submovement-triggered averages (SmTAs) of LFPs exhibited multiphasic potentials around the time of movement, as well as a second feature following submovements with a latency that depended on extrinsic delay (*Figure 8A*, *Figure 8—figure supplement 1*). This feature was revealed more clearly by reducing the dimensionality of the LFPs with PCA (*Figure 8B*). Note that if submovements reflect interference between stochastic motor errors and feedback corrections, a submovement in the positive direction can arise from two underlying causes. First, it may be a positive correction to a preceding negative error. In this case, cortical activity associated with the feedback correction should occur around time zero. Second, the submovement may itself be a positive error which is followed by a negative correction, and the associated cortical activity will hence be delayed by the feedback latency. Since the SmTA pools submovements arising from both causes, this accounts for two features with opposite polarity separated by the feedback delay. Note also that SmTAs of cursor velocity similarly overlay (negative) tracking errors preceding (positive) feedback corrections, and (negative) feedback corrections following (positive) tracking errors, evident as symmetrical troughs on either side of the central submovement peak (*Figure 8C*).

Importantly however, LFP oscillations around the time of submovements appeared largely unaffected by delay. To visualize this, we projected the SmTAs of multichannel LFPs onto the same PC plane. For all delay conditions, LFPs traced a single cycle with the same directional of rotation and comparable angular velocity (*Figure 8D*). The period of these cycles (approx. 300 ms) matched the frequency of imaginary coherence between LFPs (approx. 3 Hz). This is as expected, since signals with a consistent phase difference will be orthogonalized by PCA and appear as cyclical trajectories in the PC plane. In other words, although the precise frequency of submovements depended on extrinsic delays in visual feedback, the constant frequency of associated LFP cycles revealed delay-independent intrinsic dynamics within motor cortex. Note also that the resonant frequency of these dynamics matched the delay-independent filtering of feedback responses observed in our human experiments.

## Modelling submovement-related LFP cycles and delta oscillations in sleep

These various observations could be understood using the same computational model that explained our human behavioral data (*Figure 9*). For simplicity, we simulated two out-of-phase components within the LFP by using the total synaptic input to each of the two neural populations in the state estimator. We also added common low-frequency background noise to represent volume

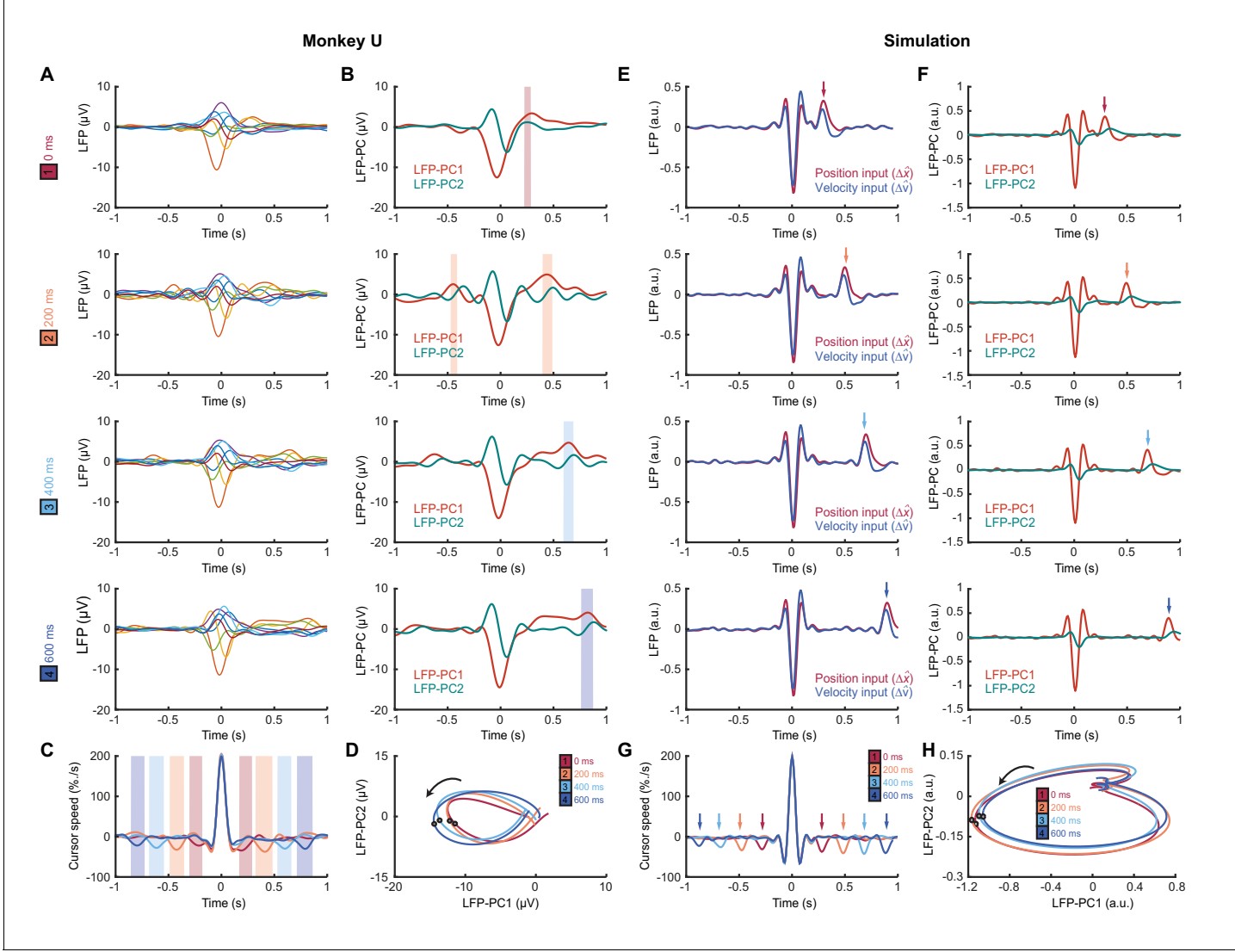

**Figure 8.** Submovement-triggered averages of M1 LFPs. (A) Average low-pass filtered LFPs from M1, aligned to the peak speed of submovements with 0–600 ms feedback delay. Note the second feature, which follows submovements by an extrinsic, delay-dependent latency. Data from Monkey U. See also *Figure 8—figure supplement 1*. (B) Average of first two LFP-PCs aligned to submovements. Shading indicates significant delay-dependent peaks in PC1 (p<0.001, Kruskal-Wallis test and post-hoc signed-ranks test across delay conditions). (C) Average low-pass filtered cursor speed, aligned to submovements. Shading indicates significant (p<0.001) delay-dependent troughs. (D) Average submovement-triggered LFP-PC trajectories, plotted over 200 ms either side of the time of peak submovement speed (indicated by circles). (E–H) Simulated submovement-triggered averages produced by the OFC model.

DOI: https://doi.org/10.7554/eLife.40145.018

The following figure supplement is available for figure 8:

**Figure supplement 1.** Submovement-triggered averages of M1 LFPs for Monkey S.

DOI: https://doi.org/10.7554/eLife.40145.019

conduction from distant sources. The simulated LFPs exhibited a broad, delay-independent spectral peak arising from the dynamics of the recurrent network (*Figure 7J*). By contrast, the resultant cursor velocity comprised the summation of motor noise and (delayed) feedback corrections, and therefore contained sharper, delay-dependent spectral peaks, due to constructive/destructive interference (*Figure 7I*). Note however, that coherence was nonetheless observed between LFPs and cursor velocity (*Figure 7K*). Time-domain SmTAs of the simulated data also reproduced features of the experimental recordings, including delay-dependent peaks/troughs reflecting extrinsic feedback delays (*Figure 8E–G*). Meanwhile, the coupling of simulated neural populations, according to

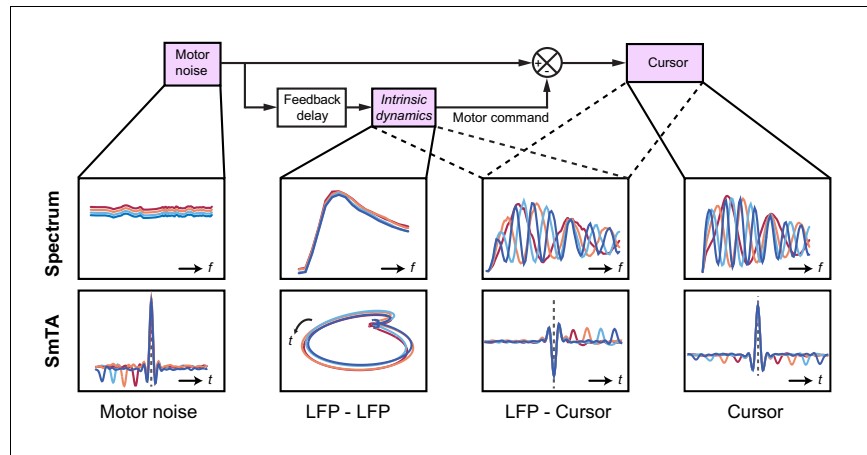

**Figure 9.** Schematic of delay-dependent and delay-independent relationships in the OFC model. The boxes show how the various frequency-domain and submovement-triggered average (SmTA) relationships are explained by the OFC model. *Top row,* from left to right: Broad spectrum motor noise drives intrinsic dynamics resulting in a delay-independent LFP cross-spectral resonance. The delayed motor command is combined with the original motor noise leading to delay-dependent comb filtering, evident in LFP-Cursor coherence and Cursor power spectrum. *Bottom row,* from left to right: submovements can arise from a positive noise peak at time-zero, or as a correction to a preceding negative noise trough. Due to intrinsic dynamics, LFPs trace consistent cyclical trajectories locked to submovements. SmTA of LFPs contains potentials associated with noise peak/troughs after feedback delay. SmTA of cursor velocity combines noise with delayed feedback corrections to yield a central submovement flanked by symmetrical troughs.
DOI: https://doi.org/10.7554/eLife.40145.020

conserved intrinsic dynamics, resulted in consistent LFP cycles around the time of movement (*Figure 8H*), and an imaginary cross-spectrum with a single delay-independent resonant peak (*Figure 7L*).

Finally, we examined whether the model could also account for cortical oscillations in the absence of behavior. Previously we have described a common dynamical structure within both cortical cycles during movement and low-frequency oscillations during sleep and sedation (*Hall et al., 2014*). In particular, K-complex events under ketamine sedation (*Figure 10A*), thought to reflect transitions between down- and up-states of the cortex, are associated with brief bursts of delta oscillation (*Figure 10B*) (*Amzica and Steriade, 1997*). The relative phases of multichannel LFPs aligned to these events matches those seen during submovements (*Figure 10D,E*). As a result, when projected onto the PC plane, LFPs trace similar cycles during both K-complexes (*Figure 10C*) and submovements (*Figure 10F*). We modelled the sedated condition by disconnecting motor and sensory connections between the feedback controller and the external world; instead providing a pulsatile input to the state estimator simulating a down- to up-state transition (*Figure 10G*). Effectively, transient excitation of the state estimator elicited an impulse response reflecting its intrinsic dynamics. The simulated LFPs generated a burst of delta-frequency oscillation around the K-complex (*Figure 10H*) which resembled submovement-related activity (*Figure 10J,K*). Projecting this activity onto the same PC plane revealed consistent cycles during simulated K-complexes (*Figure 10I*) and submovements (*Figure 10L*). Thus it appears that our computational model, incorporating the intrinsic dynamics of motor cortical networks, could also account for the conserved structure of low-frequency LFPs during movement and delta oscillations in sleep.

## Discussion

### Submovement kinematics are influenced by both extrinsic and intrinsic dynamics

Previous theories of intermittency have focused on either extrinsic or intrinsic explanations for the regularity of submovements, but little consensus has emerged over this fundamental feature of

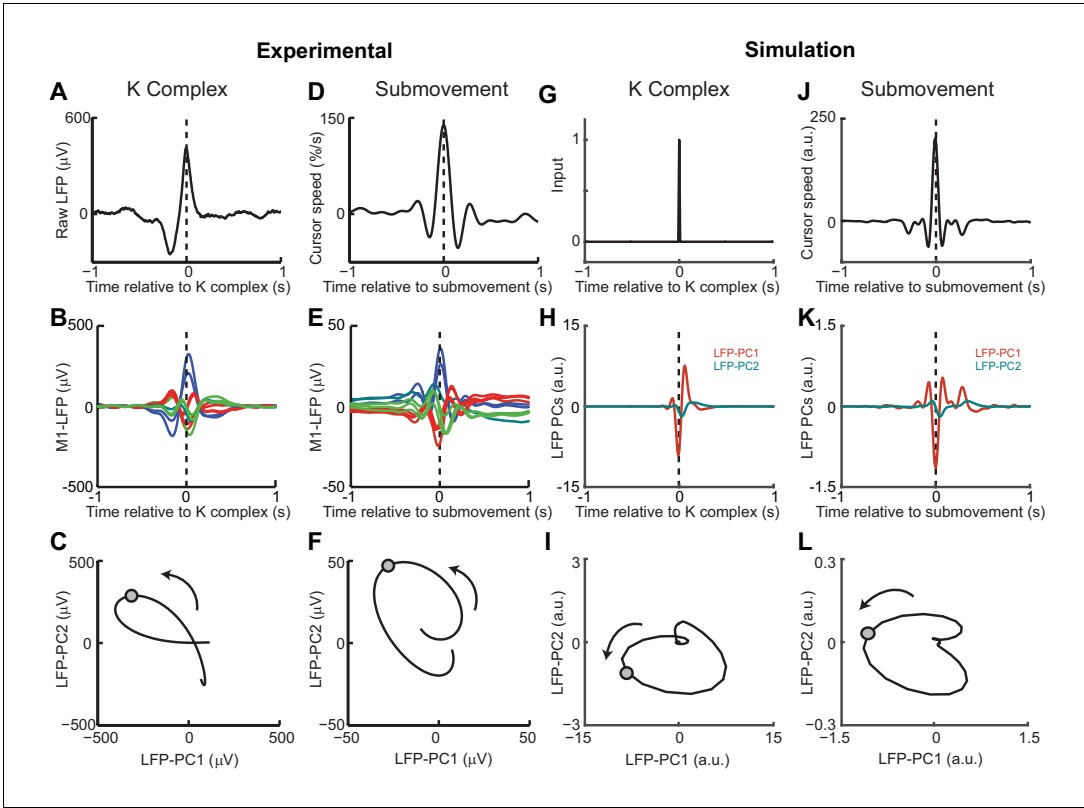

**Figure 10.** Simulated LFP dynamics during movement and sedation. (**A**) K-complex events in LFP from M1 recorded under ketamine sedation. (**B**) Average low-pass filtered multichannel LFPs aligned to K-complex events. LFPs are color-coded according to phase relative to submovements, but exhibit a similar pattern relative to K-complexes. (**C**) Average LFP-PC trajectories aligned to K-complexes, plotted over 200 ms either side of the time of the K-complex (indicated by a circle), using the PC plane calculated from recordings during awake behavior. (**D**) Average cursor speed aligned to the peak speed of submovements. (**E**) Average low-pass filtered multichannel LFPs aligned to submovements. (**F**) Average submovement-triggered LFP-PC trajectories, plotted over 200 ms either side of the time of submovements (indicated by a circle). (**G**) A K-complex under sedation is simulated by an impulse excitation of the OFC model, without connection to the external world. (**H**) Impulse response of the simulated LFP-PCs. (**I**) LFP-PC trajectories associated with simulated K-complexes. (**J**) Simulated submovement-triggered average cursor speed from the OFC model with no feedback delay. (**K**) Simulated submovement-triggered average LFP-PCs. (**L**) Simulated submovement-triggered LFP-PC trajectories. Panels A–F reproduced from Figure 4A,C,E in *Hall et al. (2014)* (published under a Creative Commons CC BY 3.0 license).
DOI: https://doi.org/10.7554/eLife.40145.021

movement. There is good evidence for a common low-frequency oscillatory structure to motor cortex activity across multiple behavioral states (*Churchland et al., 2012*; *Hall et al., 2014*; *Russo et al., 2018*) but also an influence of feedback delays on submovement timing (*Miall, 1996*). Experimentally manipulating visual feedback with artificial time delays and spatial perturbations allowed us to dissociate both contributions to submovement kinematics. We found that precise frequencies of submovement peaks were determined by delays in the extrinsic feedback loop, but that these were embedded within a delay-independent envelope reflecting intrinsic filtering of feedback corrections. This dissociation of extrinsic and intrinsic dynamics was also evident in intracortically-recorded LFPs during tracking movements. Both delay-dependent feedback corrections and delay-independent cycles were observed in submovement-triggered averages of LFPs. Moreover, while coherence between LFPs and cursor movement exhibited delay-dependent spectral peaks, the imaginary coherence between multichannel LFPs revealed a consistent dynamical structure across behaviors.

## Modelling isometric visuomotor tracking

We were able to explain these apparently contradictory results by using a continuous feedback control model, which incorporated optimal state estimation based on a second-order internal model of the external dynamics. Previously, intermittency has been implemented in optimal feedback control models by explicitly including a refractory period between submovements (*Gawthrop et al., 2011*; *Sakaguchi et al., 2015*), but theoretical justification for such an additions is lacking. In our model, submovements instead arose from constructive interference between motor errors and continuous, delayed feedback corrections. Optimal state estimation used a steady-state Kalman filter to separate process (motor) noise from measurement (sensory) noise. One free parameter was tuned to achieve correspondence between simulated and experimental data, namely the ratio of process to measurement noise, which determined the intrinsic resonance frequency around 2–3 Hz. It would be interesting in future to vary these noise characteristics experimentally (e.g. by artificially degrading visual acuity, or by extensively training subjects to produce faster or more accurate movements) and examine the effect on perturbation responses. One possible outcome would be a change to the observed resonance, although this seems to contradict the ubiquity of 2–3 Hz cortical dynamics. Alternatively, there may be other computational advantages to maintaining a consistent cortical rhythm. For example, it is notable that 2–3 Hz intrinsic dynamics matched the frequency of the primary submovement peak under unperturbed external feedback conditions, thus accentuating the fundamental submovement frequency around 2 Hz, while suppressing higher harmonics. This may be beneficial in allowing other aspects of the visuomotor machinery to be synchronized to a single rhythm, for example eye movements, which are influenced by hand movement during tracking tasks (*Koken and Erkelens, 1992*).

One puzzling feature of our results was that force amplitude responses to cursor perturbations were uniformly less than unity, which initially appears suboptimal for rejecting even slow perturbations. We first considered that proprioceptive information (which is in conflict with vision during cursor perturbations) might cause subjects to underestimate the true displacement of the cursor. However, sub-unity amplitude responses were also observed in separate experiments (not shown) when sinusoidal displacements were added to the target position. In this situation there was no discrepancy between vision and proprioception, yet subjects consistently undershot corrections to all but the lowest frequency perturbations (even in the absence of any delay). An alternative explanation is that subjects avoided making corrections requiring large changes to the motor command. This can be formalized by a cost function that is minimized by proportional-integral (PI) control, which has been used in the past to model human movement (*Kleinman, 1974*). It is more common in optimal control models to use cost functions that penalize the absolute motor command, leading to proportional feedback policies (*Todorov and Jordan, 2002*), under the assumption that this minimizes signal-dependent noise in muscles (*Jones et al., 2002*). However, the trajectory variability observed in our isometric tracking task appeared more correlated with large changes in finger forces, rather than the absolute force magnitude (*Figure 1—figure supplement 3*). Derivative-dependent motor noise was also evident as increased variability at high frequencies in our feedforward task (*Figure 2—figure supplement 3*). Since submovements result from constructive interference between tracking errors and feedback corrections, derivative-dependent motor noise also provides a counterintuitive, but necessary, explanation for why the amplitude of submovements increases with target speed (*Figure 1—figure supplement 2*). Increased intermittency cannot be a direct consequence of faster target motion, since the frequency content of this motion is nevertheless low by comparison to submovements. Rather, faster tracking requires a larger change in the motor command, leading to increased broad-band motor noise which, after constructive interference with feedback corrections, results in more pronounced peaks at submovement frequencies.

## State estimation by motor cortical population dynamics

PCA of multichannel LFPs in monkey motor cortex revealed two underlying components, which we interpret as arising from distinct but coupled neural populations. The cyclical movement-related dynamics of these components resembled those described for M1 firing rates (*Churchland et al., 2012*), which have previously been implicated in feedforward generation of movement. Specifically, it was proposed that preparatory activity first develops along 'output-null' dimensions of the neural state space before, at movement onset, evolving via intrinsic dynamics into orthogonal 'output-

'potent' dimensions that drive muscles (*Churchland et al., 2010*). However, this purely feedforward view cannot account for our isometric tracking data, since manipulation of feedback delays dissociated delay-dependent submovements from delay-independent rotational dynamics. Instead we interpret these intrinsic dynamics as implementing a state estimator during continuous feedback control, driven by noise in motor and sensory signals. We used Newtonian dynamics to construct a simple two-dimensional state transition model based on both the cursor-target discrepancy and its first derivative. While this undoubtedly neglects the true complexity of muscle and limb biomechanics, simulations based on this plausible first approximation reproduced both the amplitude response and phase-delay to sinusoidal cursor perturbations in humans, and the population dynamics of LFP cycles in the monkey. We suggest that for discrete, fast movements to static targets, transient cursor-target discrepancies effectively provide impulse excitation to the state estimator, generating a rotational cycle in the neural space. Note that this account also offers a natural explanation of why preparatory and movement-related activity lies along distinct state-space dimensions, since the static discrepancy present during preparation is encoded differently to the changing discrepancy that exists during movement. At the same time, the lawful relationship between discrepancy and its derivative couples these dimensions within the state estimator and is evident as consistent rotational dynamics across different tasks and behavioral states.

It may seem unusual to ascribe the role of state estimation to M1, when this function is usually attributed to parietal (*Mulliken et al., 2008*) and premotor areas (where rotational dynamics have also been reported, albeit at a lower frequency [*Churchland et al., 2012*; *Hall et al., 2014*]). We suggest that the computations involved in optimal tracking behaviors are likely distributed across multiple cortical areas including (but not limited to) M1, with local circuitry reflecting multiple dynamical models of the various sensory and efference copy signals that must be integrated for accurate control. These could include the estimation of the position of moving stimuli based on noisy visual inputs (*Kwon et al., 2015*), as well the optimal integration of visual and somatosensory information, which may have different temporal delays (*Crevecoeur et al., 2016*).

An alternative explanation for consistent rotational dynamics has recently been proposed by *Russo et al. (2018)*, based on the behavior of recurrent neural networks trained to produce different feed-forward muscle patterns whilst minimizing 'tangling' between neural trajectories. It is interesting to compare this with our OFC-based interpretation, since both are motivated by the problem of maintaining accurate behavior in the presence of noise. Minimizing tangling leads to network architectures that are robust to intrinsic noise in individual neurons, while OFC focusses on optimizing movements in the face of unreliable motor commands and noisy sensory signals. Given this conceptual link, it is perhaps unsurprising if recurrent neural network approaches learn implementations of computational architectures such as Kalman filters that minimize the influence of noise on behavior. In the future, it may be productive to incorporate sensory feedback into recurrent neural network models of movement, as well as including intrinsic sources of neural noise in optimal control models. The convergence of these frameworks may further help to reveal how computational principles are implemented in the human motor system.

## Materials and methods

### Subjects

Based on pilot studies, we decided in advance to use a sample size of eight subjects in each experiment. In total, we recruited 11 adult subjects at the Institute of Neuroscience, Newcastle University. Eight subjects (three females; age 23–33; one left-handed) participated in both Experiment 1 (feedback delay) and Experiment 2 (feedback delay and spatial perturbation). Eight subjects (three females; age 23–33; all right-handed) participated in Experiment 3 (feedforward task); 6 of these subjects also participated in experiments 1 and 2. Eight subjects (three females; age 23–33; all right-handed) participated in the experiment shown in *Figure 1—figure supplement 2*; 7 out of these subjects also participated in Experiment 3. All experiments were approved by the local ethics committee at Newcastle University and performed after informed consent, which was given in accordance with the Declaration of Helsinki.

## Human tracking task

Subjects tracked a (red) target on a computer monitor by exerting bimanual, isometric, index finger forces on two sensors (FSG15N1A; Honeywell). The target underwent uniform, slow, circular motion with a pseudorandom order of clockwise and anticlockwise directions across trials. Finger forces were sampled at 50 samples/s (USB-6343; National Instruments) and mapped to (yellow) cursor position, by projecting onto two diagonal screen axes. In addition, a feedback delay ($\tau_{\mathrm{ext}}$) was interposed between force and cursor movement. The feedback delay was kept constant throughout the duration of each trial (lasting 20 s). We express screen coordinates in terms of the radius of target motion, $r_{\mathrm{target}} = 100\%$. Tracking the target rotation thus required the generation of sinusoidal motion in the range of $-100\%$ to $+100\%$, corresponding to finger forces of 0 to 3.26N, with a 90° phase-shift between each hand. At the end of each trial, subjects were given a numerical score from 0 to 1000, indicating how accurately they had tracked the target. Subjects were instructed to attempt to maximize this score, which was calculated as:

$$Score = \frac{1000}{T} \times \int_{0}^{T} \left( 1 - e^{-\frac{|r_{\mathrm{cursor}}(t) - r_{\mathrm{target}}(t)|}{\delta}} \right) dt \qquad (6)$$

where $\boldsymbol{r}_{\mathrm{cursor}}(t)$ and $\boldsymbol{r}_{\mathrm{target}}(t)$ are the 2D positions of the cursor and target respectively, and $\delta = 50\%$. Apart from the experiment shown in *Figure 1—figure supplement 2*, all experiments used a frequency of target rotation, $f_{\mathrm{target}} = 0.2$ rotations per second.

Experiment 1 used five delay conditions ($\tau_{\mathrm{ext}} = 0, 100, 200, 300,$ or 400 ms). Subjects performed a total of 70 trials, comprising 14 of each condition, presented in pseudorandom order.

For Experiment 2, spatial perturbations were added to the cursor position, as well as time delays. The perturbations were equivalent to sinusoidal modulation of the target angular velocity, but were instead added to the cursor. Expressed in polar coordinates $\boldsymbol{r} = \langle r, \angle\theta \rangle$ relative to the center of the screen, the target and cursor positions were thus given by:

$$\boldsymbol{r}_{\mathrm{target}}(t) = \langle r_{\mathrm{target}}, \ \angle\, \omega_{\mathrm{target}} t \rangle \qquad (7)$$

$$\boldsymbol{r}_{\mathrm{pert}}(t) = \left\langle r_{\mathrm{target}}, \ \angle\, \omega_{\mathrm{target}} t + \frac{\omega_{\mathrm{target}}}{\omega_{\mathrm{pert}}} \sin \omega_{\mathrm{pert}} t \right\rangle - \boldsymbol{r}_{\mathrm{target}}(t) \qquad (8)$$

$$\boldsymbol{r}_{\mathrm{cursor}}(t) = \langle r_{\mathrm{force}}(t), \angle\theta_{\mathrm{force}}(t) \rangle + \boldsymbol{r}_{\mathrm{pert}}(t) \qquad (9)$$

where $\omega_{\mathrm{target}} = 2\pi f_{\mathrm{target}}$ is the angular velocity of the target around the centre of the screen, $\omega_{\mathrm{pert}} = 2\pi f_{\mathrm{pert}}$ is the angular frequency of the perturbation, and $\langle r_{\mathrm{force}}(t), \angle\theta_{\mathrm{force}}(t) \rangle$ is the unperturbed cursor position calculated from the subject's forces at time $t - \tau_{\mathrm{ext}}$.

Although the 2D cursor position was not constrained to follow the target trajectory, we did not analyze off-trajectory deviations. For simplicity, kinematic analyses were based on the time-varying angular velocity of the cursor subtended at the center of the screen:

$$\omega_{\mathrm{cursor}}(t) = \frac{d}{dt}\theta_{\mathrm{cursor}}(t) \qquad (10)$$

For spatial perturbation experiments, we also calculated the angular velocity of the unperturbed cursor position subtended at the center of the screen:

$$\omega_{\mathrm{force}}(t) = \frac{d}{dt}\theta_{\mathrm{force}}(t) \qquad (11)$$

Note that since $r_{\mathrm{force}} \approx r_{\mathrm{target}}$, the perturbation effectively adds a sinusoidal component to the angular velocity of the cursor:

$$\omega_{\mathrm{cursor}}(t) \approx \omega_{\mathrm{force}}(t) + \omega_{\mathrm{target}}\cos\omega_{\mathrm{pert}} t \qquad (12)$$

Six different spatial perturbations ($f_{\mathrm{pert}} = 0, 1, 2, 3, 4, 5$ Hz) were combined with two feedback

delays ($\tau_{\text{ext}} = 0$, 200 ms) yielding 12 conditions. Subjects performed a total of 144 trials, comprising 12 trials per condition, presented in pseudorandom order.

## Human feedforward task

In Experiment 3, we used a unimanual isometric task in which subjects were asked to make sinusoidal forces with their right index finger. Subjects received visual feedback of the cursor, but no target was shown. Instead, subjects were shown two amplitude boundaries to move between, and the frequency of movement was cued with auditory beeps at frequencies of 1, 2, 3, 4 and 5 Hz. Subjects performed a total of 15 trials, comprising three 20 s trials per frequency condition.

## Monkey experiments

### Subjects

We used two purpose-bred female rhesus macaques (monkey S: 6 years old, 6.6 kg; monkey U: 6 years old, 8.8 kg). Animal experiments were approved by the local Animal Welfare Ethical Review Board and performed under appropriate UK Home Office licenses in accordance with the Animals (Scientific Procedures) Act 1986 (2013 revision).

### Monkey isometric tracking task

Monkeys moved a 2D computer cursor by generating isometric flexion-extension (vertical) and radial-ulnar (horizontal) torques at the wrist, measured by a 6-axis force/torque transducer (Nano25; ATI Industrial Automation). Centre-out targets were presented at 8 peripheral positions in a pseudo-random order. Targets were positioned at 70% of the distance to the screen edge (100% corresponding to torque of 0.67 Nm). The diameter of the target and cursor ranged between 14 and 36%. A successful trial required maintaining an overlap between cursor and peripheral target for 0.6 s, after which the monkeys returned the cursor to the center of the screen to receive a food reward. Visual feedback of the cursor was delayed by $\tau_{\text{ext}}$ = 0, 200, 400, 600 ms throughout separate blocks of 50–70 trials each. Monkey S performed the task with the right hand. Monkey U initially used the right hand and was then retrained for a second period of data collection with the left hand.

### LFP recording

LFPs were recorded using custom arrays of 12 moveable 50 μm diameter tungsten microwires (impedance ~200 kΩ at 1 kHz) chronically implanted in the contralateral wrist area of M1 under sevoflurane anesthesia with postoperative analgesics and antibiotics. Head-free recordings were made using unity-gain headstages followed by wide-band amplification and sampling at 24.4 kilosamples/s (System 3; Tucker-Davis Technologies). LFPs were digitally low-pass filtered at 200 Hz and recorded at 488 samples/second.

Analysis of kinematics and neural data was performed on recordings over eight sessions comprising of 56 task blocks in Monkey S (no delay: 24 blocks; 200 ms delay: 13; 400 ms delay: 13; 600 ms delay: 6), and 89 sessions comprising of 356 task blocks in Monkey U (no delay: 89; 200 ms delay: 89; 400 ms delay: 89; 600 ms delay: 89). Each task block comprised 50 (monkey S) or 70 trials (monkey U).

### Human data analysis

Spectral analysis used fast Fourier transforms (FFTs) performed on non-overlapping 512 sample-point windows (approx. 10 s) taken from the middle of each trial. Submovement peaks in the power spectra were measured after smoothing with a seven-point moving-average.

For perturbation experiments, we additionally defined two complex transfer functions $H_{\text{cursor}}$ and $H_{\text{force}}$:

$$H_{\text{cursor}}\left(i\omega_{\text{pert}}\right) = \frac{2}{\omega_{\text{target}}T}\int_0^T \omega_{\text{cursor}}(t)e^{-i\omega_{\text{pert}}t}dt \tag{13}$$

$$H_{\text{force}}\left(i\omega_{\text{pert}}\right) = \frac{2}{\omega_{\text{target}}T}\int_0^T \omega_{\text{force}}(t)e^{-i\omega_{\text{pert}}t}dt \tag{14}$$

Cursor and force amplitude responses to perturbations were calculated as the magnitude of the corresponding transfer functions, and the intrinsic phase delay of force responses was given by:

$$\tau_{\varphi}\left(i\omega_{\mathrm{pert}}\right) = -\frac{\arg\left[H_{\mathrm{force}}\left(i\omega_{\mathrm{pert}}\right)\right]}{\omega_{\mathrm{pert}}} - \tau_{\mathrm{ext}} \tag{15}$$

Additionally, tracking performance was quantified off-line using the root-mean-squared Euclidean distance between cursor and target.

## Monkey data analysis

We differentiated the magnitude of the absolute 2D torque (expressed as a percentage of the distance to the edge of the screen) to obtain the radial cursor velocity. LFP channels were subjected to visual inspection to reject noisy channels prior to mean-subtraction. For time-domain analysis, LFPs and cursor velocities were low-pass filtered at 10 Hz. Submovements were defined as a peak radial cursor speed exceeding 100 %/s. For frequency-domain analysis, we took unfiltered sections of 1024 sample points from each trial (approx. 1.5 s before to 0.5 s after the end of the peripheral hold period). We subtracted the trial-averaged profile from each section before concatenating to yield long data sections without any consistent low-frequency components related to the periodicity of the task. FFTs were calculated with overlapping Hanning windows ($2^{14}$ sample points $\approx$ 34 s; 75% overlap), from which we derived the following spectra:

Cursor power: $P_{\mathrm{Cursor}}(f) = \dfrac{\sum\limits_{m=1}^{M} F_{cursor}(f,m).F_{cursor}(f,m)^{*}}{M}$

LFP power: $P_{\mathrm{LFP}\ i}(f) = \dfrac{\sum\limits_{m=1}^{M} F_{\mathrm{LFP}\ i}(f,m).F_{\mathrm{LFP}\ i}(f,m)^{*}}{M}$

LFP-cursor coherence: $Coh_{\mathrm{LFP}\ i-\mathrm{Cursor}} = \dfrac{\left|\sum\limits_{m=1}^{M} F_{\mathrm{LFP}\ i}(f,m).F_{cursor}(f,m)^{*}\right|^{2}}{M.P_{\mathrm{Cursor}}(f).P_{\mathrm{LFP}\ i}(f)}$

LFP-LFP imaginary coherence: $Im\ Coh_{\mathrm{LFP}\ i-\mathrm{LFP}\ j} = \dfrac{\left(\mathrm{Im}\left[\sum\limits_{m=1}^{M} F_{\mathrm{LFP}\ i}(f,m).F_{\mathrm{LFP}\ j}(f,m)^{*}\right]\right)^{2}}{M.P_{\mathrm{LFP}\ i}(f).P_{\mathrm{LFP}\ j}(f)}$

where $F_{\mathrm{LFP}\ i}(f,m)$ and $F_{\mathrm{Cursor}}(f,m)$ represent Fourier coefficients at frequency $f$ and window $m = (1..M)$ from LFP channel $i$ and cursor velocity respectively. All spectra were smoothed with a 16-point Hanning window. In addition, LFP power and LFP-cursor coherence were averaged across all LFP channels, while LFP-LFP imaginary coherence was averaged over all pairs of LFPs.

## Modelling

Although both human and monkey tasks involved 2D isometric control, for simplicity we modelled only a 1D controller and assumed a one-to-one mapping from control signal, $u_k$ - 2 to position, $x_k$ - 2. We neglected target motion and designed the controller to minimize the influence of stochastic motor errors using delayed, noisy feedback of position. We set the model time step $t$ - 2= 0.01 s, intrinsic feedback delay $\tau_{\mathrm{int}}$ - 2 = 0.26 s, and the ratio of process/measurement noise $\rho$ - 2= 250 s$^{-2}$ unless otherwise stated. Steady-state Kalman gains were calculated using the function *kalman* in MATLAB, and the resultant discrete time dynamic system (*Equation (4)*) was implemented by two integrating neuronal populations representing $\widehat{x}_k$ - 2 and $\widehat{v}_k$ - 2, receiving a synaptic input on each time-step equal to:

$$\begin{bmatrix}\Delta\widehat{x}_k \\ \Delta\widehat{v}_k\end{bmatrix} = \begin{bmatrix}-K_{\mathrm{pos}} & \Delta t \\ -K_{\mathrm{vel}} & 0\end{bmatrix}\begin{bmatrix}\widehat{x}_{k-1} \\ \widehat{v}_{k-1}\end{bmatrix} + \begin{bmatrix}K_{\mathrm{pos}} \\ K_{\mathrm{vel}}\end{bmatrix}y_k \tag{16}$$

Two LFP components were simulated by normalizing $\Delta\widehat{x}_k$ and $\Delta\widehat{v}_k$ to unity variance, before adding background common noise with a $\frac{1}{f}$ spectrum.

The motor command $u_k$ was generated on each time step using the Smith Predictor architecture shown in *Figure 3*. Based on our observation that trajectory variability was maximal at times when force output was changing (*Figure 1—figure supplement 3* ), we used a linear quadratic regulator (LQR) control framework to minimize a quadratic cost function, $J$, incorporating the rate of change in motor command, $\frac{u_k}{t}$:

$$J = \sum_k \left( q x_k^2 + r \left( \frac{\Delta u_k}{\Delta t} \right)^2 \right) \tag{17}$$

For a state transition matrix in the form:

$$\begin{bmatrix} x_k \\ v_k \end{bmatrix} = \begin{bmatrix} 1 & \Delta t \\ 0 & 1 \end{bmatrix} \begin{bmatrix} x_{k-1} \\ v_{k-1} \end{bmatrix} + \begin{bmatrix} 0 \\ 1 \end{bmatrix} \frac{\Delta u_k}{\Delta t} \tag{18}$$

$J$ is minimized by a state feedback policy of the form:

$$\frac{\Delta u_k}{\Delta t} = - \begin{bmatrix} K_{\mathrm{I}} \\ K_{\mathrm{P}} \end{bmatrix} \cdot \begin{bmatrix} x_k \\ v_k \end{bmatrix} \tag{19}$$

which can be integrated to yield a PI controller:

$$\begin{aligned} u_k \ &= \sum_{j=1}^{k} \Delta u_i \\ &= -K_{\mathrm{P}} \sum_{j=1}^{k} v_k \Delta t - K_{\mathrm{I}} \sum_{j=1}^{k} x_j \Delta t \\ &= -K_{\mathrm{P}} x_k - K_{\mathrm{I}} \sum_{j=1}^{k} x_j \Delta t \end{aligned}$$

We found the proportional and integral gains $K_{\mathrm{P}}$ and $K_{\mathrm{I}}$ using the function *lqr* in MATLAB with $q = 1$ and $r = \Delta t^2$. In the full model, this controller acted on the optimal estimate of position, $\hat{z}_k$, after incorporating the delay feedback loop of the Smith Predictor. Note that the transfer function of a PI controller inside the fast feedback loop of the Smith Predictor is given by *Abe and Yamanaka (2003)*:

$$H_{\mathrm{PI}}(i\omega) = \frac{K_{\mathrm{P}} + \frac{K_{\mathrm{I}}}{i\omega}}{1 + K_{\mathrm{P}} + \frac{K_{\mathrm{I}}}{i\omega}} \tag{21}$$

which equals 1 for $\omega = 0$ but tends to $\frac{K_{\mathrm{P}}}{1 + K_{\mathrm{P}}}$ at higher frequencies. Therefore this effectively reduces the response amplitude to perturbations. The full transfer function of the intrinsic dynamics, including time-delay is given by:

$$H_{\mathrm{force}}(i\omega) = e^{-i\omega(\tau_{\mathrm{int}} + \tau_{\mathrm{ext}})} H_{\mathrm{PI}}(i\omega) . H_{y \to \hat{z}}(i\omega) \tag{22}$$

$$H_{\mathrm{cursor}}(i\omega) = 1 - H_{\mathrm{force}}(i\omega) \tag{23}$$

where $H_{y \to \hat{z}}(i\omega)$ is the transfer function of the Kalman filter relating delayed position measurement to optimal position estimate.

### Data and software availability
Datasets from all human and monkey experiments, analysis code and model associated with this work are available on Dryad doi:10.5061/dryad.53sq7kn.

## Acknowledgements
We thank Jenifer Tulip and Norman Charlton for technical assistance. This work was supported by the Indonesia Endowment Fund for Education (S-2648/LPDP.3/2014), the Medical Research Council (K501396) and the Wellcome Trust (106149).

## Additional information

### Funding

| Funder | Grant reference number | Author |
|---|---|---|
| Wellcome | 106149 | Andrew Jackson |
| Indonesia Endowment Fund for Education | S-2648/LPDP.3/2014 | Damar Susilaradeya |
| Medical Research Council | K501396 | Thomas M Hall |

The funders had no role in study design, data collection and interpretation, or the decision to submit the work for publication.

### Author contributions

Damar Susilaradeya, Conceptualization, Data curation, Software, Formal analysis, Funding acquisition, Investigation, Writing—original draft, Writing—review and editing; Wei Xu, Thomas M Hall, Data curation, Investigation, Writing—review and editing; Ferran Galán, Supervision, Methodology, Writing—review and editing; Kai Alter, Conceptualization, Resources, Supervision, Methodology, Writing—review and editing; Andrew Jackson, Conceptualization, Software, Supervision, Funding acquisition, Methodology, Writing—original draft, Project administration, Writing—review and editing

### Author ORCIDs

Damar Susilaradeya (iD) http://orcid.org/0000-0002-4548-5924
Thomas M Hall (iD) http://orcid.org/0000-0002-5116-8490
Ferran Galán (iD) https://orcid.org/0000-0003-2000-5872
Andrew Jackson (iD) http://orcid.org/0000-0001-8701-6387

### Ethics

Human subjects: All experiments were approved by the local ethics committee at Newcastle University (000023/2008) and performed after informed consent, which was given in accordance with the Declaration of Helsinki.

Animal experimentation: Animal experiments were approved by the local Animal Welfare Ethical Review Board and performed under appropriate UK Home Office licenses (PPL 60/4410) in accordance with the Animals (Scientific Procedures) Act 1986. Surgeries were performed under sevoflurane anesthesia with postoperative analgesics and antibiotics, and every effort was made to reduce suffering.

### Decision letter and Author response

Decision letter https://doi.org/10.7554/eLife.40145.027
Author response https://doi.org/10.7554/eLife.40145.028

## Additional files

### Supplementary files

• Source code 1. MATLAB implementation of feedback controller model. Code used to generate *Figure 4*.
DOI: https://doi.org/10.7554/eLife.40145.022

• Transparent reporting form
DOI: https://doi.org/10.7554/eLife.40145.023

### Data availability

Datasets from the human and monkey experiments, together with sample analysis code and modelling associated with this work are available on Dryad doi:10.5061/dryad.53sq7kn.

The following dataset was generated:

| Author(s) | Year | Dataset title | Dataset URL | Database and Identifier |
|---|---|---|---|---|
| Susilaradeya D, Xu W, Hall TM, Galan F | 2018 | Data from: Extrinsic and Intrinsic Dynamics in Movement Intermittency | https://dx.doi.org/10.5061/dryad.53sq7kn | Dryad Digital Repository, 10.5061/dryad.53sq7kn |

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
