## [Decision Letter]

[**Editorial note:** This article has been through an editorial process in which the authors decide how to respond to the issues raised during peer review. The Reviewing Editor's assessment is that all the issues have been addressed.]

Thank you for submitting your article "Extrinsic and Intrinsic Dynamics in Movement Intermittency" for consideration by *eLife*. Your article has been reviewed by two peer reviewers, and by Eilon Vaadia as the Reviewing Editor. The evaluation has been overseen by Richard Ivry as the Senior Editor. One of the reviewers, Mark M Churchland (Reviewer #2), has agreed to reveal his identity.

The Reviewing Editor has highlighted the concerns that require revision and/or responses, and we have included the separate reviews below for your consideration. If you have any questions, please do not hesitate to contact us.

Summary:

The study addresses fundamental questions in motor control, using experiments (in human and monkey) and modeling (OFC) to study the phenomenon of sub-movements during bimanual isometric tracking in a circular path. The authors suggest that the intermittency observed in these continuous tracking movements is explained by an interplay of both intrinsic and extrinsic dynamics. Unlike previous studies, this work conceives of sub-movements that are observed during tracking, not as discrete entities requiring a specialized explanation, but as an interplay of both intrinsic and extrinsic dynamics, emerging as a natural consequence of a neuronal system that implements a predictive and optimal feedback control mechanism. Moreover, under conditions of delayed feedback a classic method for probing feedback control systems – sub-movement frequency content exhibits the expected shift to lower frequencies. Additional properties – the appearance of higher harmonics and a broad band-passed property – are also potentially consistent with certain kinds of continuous feedback and internal dynamics. The link to physiology is tentative but reasonable, plausible, and intriguing. For example, the finding that LFP-oscillation frequency is not delay-dependent is both interesting on its own and interconnects with behavioral observation. The overall impression of the reviewers and the reviewing editor is quite positive. However, we all feel that the paper can improve dramatically by addressing the reviewers’ concerns.

Major concerns:

See the detailed list of major and minor concerns of the two reviewers below. They strongly advise revising the style, the structure and the clarity of all sections. They also have some specific concerns. I join the reviewers' comments and encourage the authors to address all these concerns, by revising the current version of the manuscript. In particular, please note the specific comments about the modeling work.

I (E.V.) also suggest additional and special attention to the "delay perturbation" concerns. In addition to the comments made by the reviewers, I also suggest extending the short discussion of the so-called "delay-independent perturbation responses". If possible and doable, I suggest dedicating a set of experiments (human) to test adaptation to delay in your task. The first paragraph in subsection “Modelling isometric visuomotor tracking” of the discussion is problematic. My intuition fails to understand why you did not see adaptation to predictable sinusoidal perturbation. (see Sakaguchi et al., 2015) for an example a modeling work. The sentence "it would again be interesting to examine whether state estimator dynamics might adapt on a slower time-scale after extensive training with delayed feedback" suggest that the authors understand the problematics. How long is the longer timescale. The present form of the paper suggests that it was not tested. I see this as a serious drawback of an otherwise elegant study.

Separate reviews (please respond to each point):

*Reviewer #1:*

This paper by Susilaradeya and colleagues attempts to determine the cause of sub-movements. The place their theoretical formulation within the context of optimal feedback control and test their ideas both in human and non-human primates, both at the level of behavior and neurophysiology. This is an interesting paper. In the spirit of this review pilot, I have no comments that require essential revisions in terms of data analysis or data collection so I leave these kinds of quibbles out, but I do have some major comments in terms of conceptualization that the authors may want to consider to improve the accessibility of their ideas.

1) The Introduction is meandering and it doesn't present the logic of the present investigation in a clear enough way. This leaves the reader hanging when they get to the Results (see #3 below). I think the authors need to get to the point of the intrinsic/extrinsic interactions and optimal feedback control more quickly and make it clear what gap they are filling.

2) The sleep bits seem completely out of place until it appears near the end of the discussion. I would seriously consider dropping its mention from earlier in the manuscript because no one will have any idea what is going on anyway.

3) The results are tough to take without more narrative at the beginning of each analysis section. I think this is particularly difficult with the first section of the results because everything is coming out of nowhere. Eventually the rationale is mopped up at the end of the section but by then its too late. Lead up front with some motivation of why you are doing each of these things.

4) The authors use the intervention of increasing visual delays and visual perturbations as a means for probing the state of the system. How does this limit the generalizability of their findings? There is, for example, the following paper that describes the tradeoff of using fast but crude somatosensory inputs versus slow but acute visual ones. Might be worth considering more deeply (477-481) the interplay between many modalities in terms of the OFC framework. http://www.jneurosci.org/content/jneuro/36/33/8598.full.pdf

Minor Comments:

1) The figures are very small, it was very hard to appreciate any details in these. They need to be substantially bigger in print.

*Reviewer #2:*

Overall, I found this to be a strong study with compelling results and a novel interpretational perspective that could be a potential leap forward in characterizing and explaining the phenomenon of sub-movements. The phenomenon of sub-movements has always been intriguing, but the relevant subfield has been contentious and has thus perhaps not received the attention it deserves (I will here admit to being relatively ignorant of that subfield). The present study conceives of sub-movements not as discrete entities requiring a specialized explanation, but as a natural consequence of feedback control. Under conditions of delayed feedback – a classic method for probing feedback control systems – sub-movement frequency content exhibits the expected shift to slower frequencies. Additional properties – the appearance of higher harmonics and a broad band-passed property – are also potentially consistent with certain kinds of continuous feedback and internal dynamics. This view struck me as both conceptually appealing and quite successful empirically. The link to physiology is tentative, but reasonable, plausible, and intriguing. For example, the finding that LFP-oscillation frequency is not delay-dependent is both interesting on its own, and meshes nicely with behavioral observations.

Yet despite my broad enthusiasm, I found the study to be a frustrating read. It took at least five times the cognitive effort, relative to a typical study, for me to fully (ok, 85%) understand everything. I read the manuscript three times and visited multiple Wikipedia pages in the process. This is despite the fact that I have some (not a lot, but some) experience with control-theory style modeling. Some of this cognitive effort is both necessary and rewarding – the authors are bringing multiple concepts to bear on both behavioral and physiological data, in a way that is motivated by the data and helps make sense of it. It is acceptable that this produces a challenging read, and I definitely learned things in the process. Yet at the same time, aspects of the presentation could be altered to make life easier (ideally much easier) on the reader. To be blunt, the manuscript reads as if it were written by a very bright graduate student who was given too much leeway in determining how the story is structured and told. The manuscript often offers trees-centric explanations where forest-centric explanations would be more appropriate. This not only impacts readability, but also results in attempts to over-explain data with specific models that, in at least one case, seem implausible.

The central findings of the study are relatively simple and fairly easy to interpret. Artificially increasing feedback delay alters the frequency content of sub-movements. Thus, sub-movements timing is not set by an internal clock, but by feedback dynamics. Indeed, the modeling and analysis demonstrate that 'sub-movements' need not be thought of as discrete entities at all – all the data are quite compatible with purely continuous control. Analysis of the response to perturbations further supports the above view. Yet at the same time, the data reveal that there also seem to be an intrinsic mechanisms that 'likes' to operate in the 1-3 Hz range. Although feedback delay alters sub-movement frequency, when frequencies fall in this band they are potentiated (equivalently, frequencies that fall outside this band are attenuated). This finding indicates that there is some intrinsic rhythmicity / band-pass aspect to the generation of movement corrections. This behavioral-level observation is linked with the observation that LFP-based oscillations are prominent in the 1-3 Hz range. Importantly, this band is not altered by changing the feedback delay. Thus, sub-movement properties are consistent with operation of feedback control (where the delay determines frequency content) and an intrinsic system (perhaps the same one that generates the LFP oscillations) that band-passes the responsiveness of the feedback-control system.

In conveying and interpreting the above results, the authors employ control-theory style models. This is important: the models help to transcend hand-waviness and give concrete examples of how a system might work and what one would expect empirically. Certain results (e.g., a peak in frequency content, the shift in that peak with imposed delay, and the appearance of higher harmonics) are actually much easier to understand with a concrete model. Given this, I would not recommend removing the modeling. Yet at the same time, the 'full' model endorsed by the authors – and repeatedly offered as an explanation for all the observed phenomena – makes at least one critical assumption that strikes me as utterly implausible.

To see this, note that in the model in Figure 3, the authors include, in the internal loop, the FULL delay (natural plus imposed). Trials with different delays are interleaved. Thus, the model assumes that, on a single trial, the nervous system can somehow infer the correct delay. This strains credulity well past the breaking point. Indeed, the only reason to make this assumption is that otherwise, the model would not work. I simulated this model (with a few simplifications) and found that even a very modest mismatch in total and internal delay caused the model to become very unstable. In contrast, behavior does not become unstable when an artificial delay is added. Thus, for one to believe the explanation embodied in this model, one would have to imagine that the brain infers the correct delay almost instantaneously and very accurately. This is hard to swallow. The authors note the importance of this assumption in the Discussion, but do not seem to adequately appreciate how implausible it is. Are we really to believe that the cerebellum can, in under a second, correctly infer the exact delay on each trial? It is possible that I am missing some key fact here, but if not, this specific model really seems like a non-starter.

Related (but much less troubling) points can be made regarding the model element that captures the intrinsic band-passed property. For the model to work, it is critical (I think) that process noise impact not force per se, but the derivative of force (i.e., cursor velocity). This runs contrary to the most natural modeling assumption, and the one typically made: that force is corrupted by motor noise. By assuming that force is corrupted by the INTEGRAL of motor noise, the authors are basically building in the band-passed feature (high-frequency errors should be filtered out). This seems a bit like cooking the books after knowing the result. This seems unnecessary. The most important aspects of the interpretation don't depend critically on the exact model – simply on there being some internal process with appropriate band-pass properties. The authors themselves note this in the discussion. Thus, some balance needs to be struck between using a concrete model to illustrate a broader point, and not forcing the reader to dig deep to understand a specific model that is rather post-hoc and is only one of many plausible models.

I believe it likely that the authors can revise the manuscript to address these shortcomings. More of the focus should be on the general properties of the data and what they imply at a broad level. Control-theory-style models can (and probably should) be used to illustrate fundamental properties, but less should be made of specific pet models. This is especially true when the conceptual point is a fairly general one, and when many related models might show the key properties. This would also allow the manuscript to be more readable. The manuscript could better indicate which details really matter and which don't. Then the reader would know when to pause and really understand a point, and when to read on.

Along those lines, the manuscript needs to do a better job of helping the reader at key moments. When key concepts come up, we may need a bit of explanation (e.g., the interpretation of the cross-spectrum is pretty simple, but most readers will need a sentence to help them along). I would suggest having peers from slightly different fields read the manuscript and flag the junctures where they struggled and got bogged down. I am pretty close to the center of mass of the intended audience, and I struggled at times. In the end I found the struggle worth it, but would still have preferred a cleaner telling of the story.

Detailed comments:

1) I simulated the authors' model from Figure 3, and also a more old-fashioned control-theory style model. This second model had no explicit internal predictor but simply responded to the error and the derivative in the error. This is an old strategy which basically amounts to predicting the present error from the first two terms in the Taylor series. As has been done in many models from the oculomotor system, I incorporated non-linearities on both the error and derivative-of-error signals. This aids stability, and incidentally causes the impulse response to contain harmonics. I assumed a feedback delay of 150 ms – a more empirically defensible estimate of the delay during movement than 300 ms estimate used by the authors. This model did an adequate job of reproducing the first key result of the study. I don't wish to argue that the model I simulated is a better model overall. It surely has some deficiencies. Rather, this exercise makes the point that there are multiple related models that could work. The key point is that they are all feedback control models where the frequency content shifts leftwards with longer delays. Thus, the data argue (pretty unambiguously it seems) for some model from this class. This important broader point should not be lost in the details of particular models.

2) The term 'Optimal Control Model' is used often. I agree that the Kalman filter is an optimal estimator of the state. I'm not so sure about the whole model. I generally think of 'Optimal Feedback Control' (OFC) models as having a feedback law that is optimized based on a particular cost function. Typically that feedback law is time-varying and specific to each action (e.g., it is different when reaching right vs left). Perhaps the model presented is optimal in the case of a very simple cost function (keep the output near zero at all times). I'm not sure. I thus ask the authors to think carefully regarding whether the model is really an OFC model, or merely a good feedback controller that employs optimal state estimation.

3) I found Figure 1F confusing when first described. It isn't a feedback system, but is described as such. Only later do we learn that certain types of feedback control systems can be formally reduced to the diagram in 1F. This is one of those places where the cognitive load on the reader spikes.

4) Speaking of cognitive load, I initially found the third row of Figure 2 (panels C,D,E) rather confusing. It isn't immediately obvious exactly what is different from the above row. Even once one figures that out, the summary results in panel H seem not to obviously agree with the results in panels F and G. E.g., the peak at 2 Hz is much higher in panel F than G, yet this isn't reflected in panel H. I think I understand this in the end (after subtracting baseline and taking the square root, the difference is pretty small), but this threw me for a while.

There are other instances where I had to struggle a bit to understand figure panels. In the end I was typically able to (and satisfied when I did) but I felt I shouldn't have had to work so hard.

5) The Kalman-based explanation for the band-pass property is a reasonable one, but there are other reasonable explanations as well. This is appropriately handled in the Discussion. Yet this broader perspective comes rather late. This caught my eye because the authors connect their results (both behavioral and LFP-based) with recent findings that show ~2 Hz quasi-oscillatory dynamics in motor cortex. That is indeed a plausible connection, but not obviously consistent with the Kalman-filter-based explanation. For brisk reaches, plotting cursor position vs cursor velocity would produce an oscillation much higher-frequency than 2 Hz. Instead, it is the muscle activity that has ~2 Hz features during reaching. This is perhaps consistent with the Kalman idea, but at that point the quantity being filtered would be muscle activity not cursor position. I don't think these facts detract from the authors' findings, but they do suggest that interpretation could benefit if it were less tied to a very specific model.

6) The authors explain the band-passed property of their Kalman filter as follows: "However, for movements in the physical world, it is unlikely that high-frequency tracking discrepancies reflect genuine motor errors, since this would imply implausibly large accelerations of the body." The idea is that the feedback control system should tend to ignore high-frequency errors in the visual feedback, on the grounds that they are too high frequency to have come from the plant, and therefore don't need correcting.

This seems at odds with their previous argument that the plant "can generate force fluctuations up to 5 Hz with little attenuation (Figure 2J,K and Figure 2—figure supplement 3)." If so, then frequencies should be ignored only above 5 Hz; frequencies this high absolutely could be due to motor noise and should not be ignored.

This was another juncture where I found I had to think unnecessarily hard about the specifics of the authors model, when really I'd rather be thinking about broader interpretations.

7) On a related note, the Kalman-filter-based explanation for 'intrinsic filtering' assumes (if I understand correctly) that process noise impacts cursor velocity directly, yet impacts cursor position only through integration. This is critical as it is this fact that makes high-frequency fluctuations of cursor position something that should be filtered out. If process noise directly impacted cursor position then this would not be true and the model would not provide an explanation for the empirical band-pass property.

The assumption that noise impacts cursor velocity would make sense if the cursor were, say, the physical position of a body part. Noise at the level of force production would cause fluctuations in velocity, which would integrate to cause fluctuations in position. However, in the present task cursor 'position' is not the position of a limb, but directly reflects force. Any noise in force production should directly impact cursor position, and need not be integrated. There seems to me no justification for assuming that process noise impacts the rate-of-change-of-force. This seems like a rather unnatural and unusual assumption. Given this, I found the overall explanation not overly compelling.

Additional data files and statistical comments:

There are certainly a few places where formal statistical tests (most likely bootstraps) could be added. For example, some of the features in Figure 7AB are quite small but potentially very revealing. Backing up their presence with statistical tests is thus appropriate.

[Editors' note: further revisions were suggested, as described below.]

We are pleased to inform you that your article, "Extrinsic and Intrinsic Dynamics in Movement Intermittency", will be published by *eLife*. We offer some comments that you should consider in preparing a final version for publication – we leave it to you to decide what changes to be made. Also, given this is a peer review paper, the final decision to publish is yours.

This study is a compelling investigation of the phenomenon of submovements, bringing together multiple experimental approaches to produce a novel perspective. The study views submovements not as discrete entities, but the natural product of feedback control that cancels errors at some frequencies but not others. This account might seem to also predict that internal correlates of rhythmic submovements should also become slower with longer delays. Interestingly this doesn't happen. The authors relate this physiological finding to an experimental finding; Although the frequencies observed in behavior are delay-dependent, there appears to be a delay-independent windowing with a peak at 2-3 Hz. That is, the system seems to 'prefer' to operate around 2-3 Hz, and frequencies in that range are relatively accentuated. This 2-3 Hz frequency agrees with what is observed in the LFP. Enticingly, prior work has also found a prominent 2-3 Hz frequency in neural activity during rhythmic and non-rhythmic tasks. The authors present a unified model, involving internal filtering (favoring 2-3 Hz) within a feedback loop. This model successfully explains the major features of the data.

The first submission of this manuscript was a bit of a tough read (all reviewers were of this opinion). We find the revised version much improved. One reviewer does offer some minor revisions. His comments follow:

"I really only have comments regarding changes to improve readability… Even though a couple of my comments are lengthy, I have labelled all these comments as 'minor', as I feel confident they can be readily addressed. I trust the authors to digest these points and to figure out what changes should be made.

Minor comments:

The authors use the new Figure 5 to address a criticism raised in the last round. While their model does a nice job of accounting for the data, it makes the potentially implausible assumption that the internal delay (in the Smith predictor) is able to rapidly update itself to reflect the sum of internal and imposed delays. The concern is that, because delays are different on different trials, it seems implausible that internal delay would always match the imposed delay. To address this concern, the authors point out that the brain has more opportunity to adapt than we had initially supposed: subjects were able to 'get used' to the delay for 5 seconds before the section of the data being analyzed. Furthermore, the new Figure 5 demonstrates that some kind of adaptation does occur during this time: models fit to the data need different parameters for the first 5 seconds versus the last 5.

I still have my doubts that 5 seconds is enough time to recalibrate an internal estimate of the feedback delay. That said, other aspects of the model are very successful, and in general, I think the results go beyond any particular model. The paper would still be interesting and the take-home messages similar even if a somewhat different feedback model had to be proposed. For these reasons, I am largely satisfied, at a scientific level, with the way in which this concern has been addressed. That said, I found the newly added section to be confusing in multiple ways.

First, the way the section is introduced doesn't help the reader understand why the section (and the large accompanying figure) is there. The sentence beginning with 'However, since…' is helpful only if you already have thought about this problem. I would suggest that the section begin by confronting the problem head-on, and clearly stating that a challenge for the model is that it would have to adapt very rapidly. Therefore it is worth asking whether some form of rapid adaptation occurs. Fully understanding the motivating concern would help the reader understand why this section is here (and allow them to make up their own mind regarding how convinced they are).

Other aspects of this new section were also confusing to me. Subsection “Emergence of delay-specific predictive control during individual trials” states 'We asked which model could better explain the experimentally-observed force amplitude responses by inferring the associated intrinsic dynamics under both architectures.' This left me expecting more plots like those in Figure 4. But the analyses in Figure 5 don't (I think) really address this issue. Rather, they show that some kind of adaptation occurs. Given that it does, we have less reason to doubt a central modeling assumption (after all, any model would have to adapt in some way). This seems a reasonable argument, but that isn't how the section was motivated or set up.

In this context, I also found the plots in E and F to be a bit confusing. They are plots of the gains used to fit the data, not plots of gains produced by the model. The plots in F seem to be a better match to the data, which at first seems like the point of the plots. However, I think that is actually a red herring. The main point (I think) is just that some kind of adaptation has to be assumed, which addresses the concern that fast adaptation is implausible.

This section should be carefully rewritten to be more explicit regarding why the analysis was performed, how to interpret the plots, and what exactly is being concluded.

For Figure 1E, where did lines come from? Regression? This seems to be the case (it is implied by the legend of Table 1) but the actual figure legend doesn't explicitly state this and I don't think the Results do either.

Along similar lines, why not add the lines predicted by the model to Figure 1E? The info is currently presented in Table 1, but would likely be more than compelling if added to this key plot. It would be nice to see that the slopes and intercepts match fairly well (it needn't be perfect of course).

I liked the “Outline” section but the heading “Outline” is perhaps a bit heavy handed?

My impression was that the cursor is not constrained to live on the circle. E.g., there could be radial errors (unanalyzed here). If so that should be stated explicitly in the Materials and methods (my apologies if I missed this).

Subsection “Intrinsic dynamics in the visuomotor feedback loop” states 'Analyzed in this way, amplitude responses were largely independent of extrinsic delay.' When I first read this, I found it all rather confusing. Analyzing different ways gives different results? Which is correct? Is there some sort of conflict? It all makes sense in the end, but a bit of help at this critical juncture might save someone some temporary confusion.

Regarding Figure 4J. This plot has the potential to generate some confusion, as it is actually data, but presented in the context of a figure where nearly all the other panels (A-I) contain only model behavior. Also, there is a dashed line that seems like it might be a model prediction, but might in fact just be the result of a linear regression (this isn't state). This all needs to be clarified. Also, if this is an analysis of data testing a model prediction, then the model prediction should be shown. Otherwise we are comparing to a prediction that isn't illustrated.

Subsection “Intrinsic cortical dynamics are unaffected by feedback delays” paragraph four: 'the two neural population<s>'

---

## [Author Response]

We thank the reviewers and reviewing editors for their time in considering our manuscript. We are pleased that the reviewers felt our study ‘addresses fundamental questions in motor control’ (Reviewer 1) with ‘compelling results and a novel interpretational perspective’ (Reviewer 2). The reviewers also raise excellent points that have stimulated new analyses as well as clarifications to the text.

Reviewing Editor comment:See the detailed list of major and minor concerns of the two reviewers below. They strongly advise revising the style, the structure and the clarity of all sections.Reviewer #1 comment:1) The Introduction is meandering and it doesn't present the logic of the present investigation in a clear enough way. This leaves the reader hanging when they get to the Results (see #3 below). I think the authors need to get to the point of the intrinsic/extrinsic interactions and optimal feedback control more quickly and make it clear what gap they are filling.

We have shortened our Introduction in an attempt to express more clearly the logic of our study, e.g.:

“It has been proposed that the intrinsic dynamics of recurrently-connected cortical networks act as an ‘engine of movement’ responsible for internal generation and timing of the descending motor command (Churchland et al., 2012). However, another possibility is that low-frequency dynamics observed in motor cortex arise from sensorimotor feedback loops through the external environment. On the one hand, cortical cycles appear conserved across a wide range of behaviors and even share a common structure with δ oscillations during sleep (Hall et al., 2014), consistent with a purely intrinsic origin. On the other hand, the influence of feedback delays on submovement timing suggests an extrinsic contribution to movement intermittency. Therefore we examined the effect of delay perturbations during isometric visuomotor tracking in humans and monkeys to dissociate both delay-independent (intrinsic) and delay-dependent (extrinsic) components of movement kinematics and cortical dynamics.”

Reviewer #1 comment:3) The results are tough to take without more narrative at the beginning of each analysis section. I think this is particularly difficult with the first section of the results because everything is coming out of nowhere. Eventually the rationale is mopped up at the end of the section but by then its too late. Lead up front with some motivation of why you are doing each of these things.Reviewer #2 comment:Along those lines, the manuscript needs to do a better job of helping the reader at key moments. When key concepts come up, we may need a bit of explanation (e.g., the interpretation of the cross-spectrum is pretty simple, but most readers will need a sentence to help them along). I would suggest having peers from slightly different fields read the manuscript and flag the junctures where they struggled and got bogged down. I am pretty close to the center of mass of the intended audience, and I struggled at times. In the end I found the struggle worth it, but would still have preferred a cleaner telling of the story.

In revising our manuscript, we have endeavoured to improve readability of the results. For example, we have added an outline to the beginning of the Results on to indicate the overall structure:

“Our results are organized as follows. First, we describe behavioral results with human subjects, examining the effects of delay perturbations on movement intermittency and feedback responses during an isometric visuomotor tracking task. Second, we introduce a simple computational model to illustrate how principles of optimal feedback control, and in particular state estimation, can explain the key features of our data. Finally, we examine local field potentials recorded from the motor cortex of monkeys performing a similar task, to show that cyclical neural trajectories are consistent with the implementation of state estimation circuitry.”

In addition, we have added sentences referring to relevant steps in our approach to the beginning of each section to motivate the analyses contained within, for example:

“Our first experiment aimed to characterize the dependence of submovement frequencies on feedback delays…”

“According to this extrinsic interpretation, intermittency arises not from active internal generation of discrete submovement events, but as a byproduct of continuous, linear feedback control with inherent time delays. Submovement frequencies need not be present in the smooth target movement, nor do they arise from controller non-linearities. Instead these frequencies reflect components of broad-band motor noise that are exacerbated by constructive interference with delayed feedback corrections. To seek further evidence that intermittency arises from such constructive interference, we performed a second experiment in which artificial errors were generated by spatial perturbation of the cursor.”

“Although submovement frequencies depended on extrinsic feedback delays, examination of the velocity spectra in Figure 1D suggests that intermittency peaks were embedded within a broad, delay-independent low-pass envelope. This envelope could simply reflect the power spectrum of tracking errors (i.e. motor noise is dominated by low-frequency components). However, an additional possibility is that the gain of the feedback controller varies across frequencies (e.g. low-frequency noise components generate larger feedback corrections). To explore the latter directly, we examined subjects’ force responses to our artificial perturbations.”

“The amplitude and phase responses to perturbations during human visuomotor tracking provided evidence for intrinsic low-frequency dynamics in feedback corrections, which we have interpreted in the framework of optimal state estimation. The schematic on the right of Figure 3A suggests how a simple Kalman filter could be implemented by neural circuitry, with two neural populations (representing position and velocity) evolving according to Equ. 4 and exhibiting a resonant cross-spectral peak (Figure 3D). To seek further evidence for the neural implementation of such a filter we turned to intracortical recordings in non-human primates…”

Furthermore, we have expanded our discussion of the cross-spectrum:

“Figure 3B,C shows the amplitude response for position and velocity estimates produced by the Kalman filter. Since these are out of phase with each other, their cross-spectral density (which captures the amplitude and phase-difference between frequency components common to both signals) will generally be complex. Broadband input therefore results in an imaginary component to this cross-spectrum with a characteristic low-frequency resonance peak determined by the state estimator dynamics (Figure 3D).”

“Finally, we calculated imaginary coherence spectra between pairs of LFPs (see Materials and methods). The imaginary component of coherence indicates frequencies at which there is consistent out-of-phase coupling in the LFP cross-spectrum. This effectively separates locally-varying oscillatory components from in-phase background signals (e.g. due to volume conduction from distant sources), and revealed more clearly the 2-3 Hz LFP oscillation (Figure 7D,H).”

Evidence for Smith Predictor and time-course of adaptation:

Reviewer #2 and the Reviewing Editor both had concerns about the computational model, and whether there was evidence for delay-dependent adaptation during the course of the trial. We have addressed these concerns with an additional analysis and a new Figure 5. We have grouped the reviewers’ comments below, in order to best explain how our new analyses address all these concerns.

Reviewer #2 comment:Yet at the same time, the 'full' model endorsed by the authors – and repeatedly offered as an explanation for all the observed phenomena – makes at least one critical assumption that strikes me as utterly implausible.To see this, note that in the model in Figure 3, the authors include, in the internal loop, the FULL delay (natural plus imposed). Trials with different delays are interleaved. Thus, the model assumes that, on a single trial, the nervous system can somehow infer the correct delay. This strains credulity well past the breaking point. Indeed, the only reason to make this assumption is that otherwise, the model would not work. I simulated this model (with a few simplifications) and found that even a very modest mismatch in total and internal delay caused the model to become very unstable. In contrast, behavior does not become unstable when an artificial delay is added. Thus, for one to believe the explanation embodied in this model, one would have to imagine that the brain infers the correct delay almost instantaneously and very accurately. This is hard to swallow. The authors note the importance of this assumption in the Discussion, but do not seem to adequately appreciate how implausible it is. Are we really to believe that the cerebellum can, in under a second, correctly infer the exact delay on each trial? It is possible that I am missing some key fact here, but if not, this specific model really seems like a non-starter.

First, we apologise that an important piece of information needed to interpret our analyses was buried in the Materials and methods section of the previous submission. Since it was not our original intention to study adaptive processes, we focussed only a 10s window in the middle of each trial (from 5-15s after trial start). Therefore subjects have at least 5 s to adjust to the new delay before the analysis window starts. We now state this important information multiple times in the text of the Results section, as well as in the legends of the relevant figures. We did not feel that the inclusion of the internal efference copy loop when interpreting our data was particularly controversial as it is often included in OFC-type models. However, the reviewer raises an interesting point as to whether it is actually justified by our data, and whether there is evidence for adaptive changes through the trial, which we now address (see next point).

Reviewer #2 detailed comment:1) I simulated the authors' model from Figure 3, and also a more old-fashioned control-theory style model. This second model had no explicit internal predictor but simply responded to the error and the derivative in the error. This is an old strategy which basically amounts to predicting the present error from the first two terms in the Taylor series. As has been done in many models from the oculomotor system, I incorporated non-linearities on both the error and derivative-of-error signals. This aids stability, and incidentally causes the impulse response to contain harmonics. I assumed a feedback delay of 150 ms – a more empirically defensible estimate of the delay during movement than 300 ms estimate used by the authors. This model did an adequate job of reproducing the first key result of the study. I don't wish to argue that the model I simulated is a better model overall. It surely has some deficiencies. Rather, this exercise makes the point that there are multiple related models that could work. The key point is that they are all feedback control models where the frequency content shifts leftwards with longer delays. Thus, the data argue (pretty unambiguously it seems) for some model from this class. This important broader point should not be lost in the details of particular models.

It is worth noting that the 300 ms delay time is confirmed in our perturbation experiments, and does not seem unfeasible based on human visual reaction times. However, we agree with the general point that it is not helpful to get bogged down in the details of particular models. Our approach was to choose the simplest model compatible with our findings, while acknowledging that many more complex models may equally fit the data. However, this raises an important point – is the Smith Predictor necessary to explain our data or would a simpler non-adaptive feedback controller suffice? We now analyse this explicitly in a new Figure 5 and describe the resultant analysis in the section “Emergence of predictive control during individual trials”. We show that adaptation clearly occurs early in the trial. If a simple feedback controller model is assumed, then this adaptation appears as a rather complex pattern of sharp peaks and troughs emerging in the controller dynamics. By contrast, if a Smith Predictor is assumed, then the associated intrinsic dynamics converges to broad, delay-independent filter. We conclude that:

“Therefore adaptation culminated in a control strategy that could parsimoniously be interpreted as an appropriately-calibrated Smith Predictor with delay-independent intrinsic dynamics.”

Reviewing Editor comment:I (E.V.) also suggest additional and special attention to the "delay perturbation" concerns. In addition to the comments made by the reviewers, I also suggest extending the short discussion of the so-called "delay-independent perturbation responses". If possible and doable, I suggest dedicating a set of experiments (human) to test adaptation to delay in your task. The first paragraph in subsection “Modelling isometric visuomotor tracking” of the discussion is problematic. My intuition fails to understand why you did not see adaptation to predictable sinusoidal perturbation. (see Sakaguchi et al., 2015) for an example a modeling work. The sentence "it would again be interesting to examine whether state estimator dynamics might adapt on a slower time-scale after extensive training with delayed feedback" suggest that the authors understand the problematics. How long is the longer timescale. The present form of the paper suggests that it was not tested. I see this as a serious drawback of an otherwise elegant study.

This section of the discussion was misleading and has now been removed. It is not the case that we do not see adaptation, nor that perturbation responses are delay-independent. As outlined above, our main analyses cover a time window after most of the adaptation has taken place, and the delay-dependence can parsimoniously be explained by a simple Smith Predictor architecture with a delay-independent state estimator. However, as requested we have added new analyses to determine the time-course of adaptation. These are included in the new Figure 5 and show clear adaptation within a single trial including:

1) Force response profiles that progressively resemble those expected from a Smith Predictor with a delay-independent feedback gain and a correct internal delay model.

2) A progressive reduction in low-frequency phase delay consistent with predictive state estimation.

3) A progressive decrease in the lag between cursor and target in the presence of artificial feedback delay.

Since these time-courses appear to asymptote during the trial, we have removed speculations about longer time-course adaptation. We have added reference to the relevant Sakaguchi et al., 2015, study and thank the reviewing editor for bringing this to our attention. However, it is worth noting that this study explicitly neglects modelling of motor (process) noise. Since our study shows that submovements arise from constructive interference between motor noise and continuous feedback corrections, this may account for their modelling results suggesting that continuous control does not generate intermittency.

Justification for Kalman filterReviewer #2 comment:Related (but much less troubling) points can be made regarding the model element that captures the intrinsic band-passed property. For the model to work, it is critical (I think) that process noise impact not force per se, but the derivative of force (i.e., cursor velocity). This runs contrary to the most natural modeling assumption, and the one typically made: that force is corrupted by motor noise. By assuming that force is corrupted by the INTEGRAL of motor noise, the authors are basically building in the band-passed feature (high-frequency errors should be filtered out). This seems a bit like cooking the books after knowing the result.Reviewer #2 detailed comment:6) The authors explain the band-passed property of their Kalman filter as follows: "However, for movements in the physical world, it is unlikely that high-frequency tracking discrepancies reflect genuine motor errors, since this would imply implausibly large accelerations of the body." The idea is that the feedback control system should tend to ignore high-frequency errors in the visual feedback, on the grounds that they are too high frequency to have come from the plant, and therefore don't need correcting.This seems at odds with their previous argument that the plant "can generate force fluctuations up to 5 Hz with little attenuation (Figure 2J,K and Figure 2—figure supplement 3)." If so, then frequencies should be ignored only above 5 Hz; frequencies this high absolutely could be due to motor noise and should not be ignored.This was another juncture where I found I had to think unnecessarily hard about the specifics of the authors model, when really I'd rather be thinking about broader interpretations.7) On a related note, the Kalman-filter-based explanation for 'intrinsic filtering' assumes (if I understand correctly) that process noise impacts cursor velocity directly, yet impacts cursor position only through integration. This is critical as it is this fact that makes high-frequency fluctuations of cursor position something that should be filtered out. If process noise directly impacted cursor position then this would not be true and the model would not provide an explanation for the empirical band-pass property.The assumption that noise impacts cursor velocity would make sense if the cursor were, say, the physical position of a body part. Noise at the level of force production would cause fluctuations in velocity, which would integrate to cause fluctuations in position. However, in the present task cursor 'position' is not the position of a limb, but directly reflects force. Any noise in force production should directly impact cursor position, and need not be integrated. There seems to me no justification for assuming that process noise impacts the rate-of-change-of-force. This seems like a rather unnatural and unusual assumption. Given this, I found the overall explanation not overly compelling.

We accept that a Newtonian dynamics model is a simplifying assumption, although it is perhaps not as unreasonable as the reviewer suggests. First, the Kalman filter is estimating the discrepancy between target and cursor, and therefore includes a model of the target dynamics. Of course, the target dynamics in a neuroscience experiment could be arbitrary, but it is not implausible that subjects might expect moving targets to tend to continue moving in the same direction. Second, it is well known that noise in isometric force production is not uncorrelated, but is instead dominated by low-frequency components. Third, we show in Figure 1—figure supplement 3 that motor noise is not proportional to force level but instead to the change in force – such a result is at least consistent with some kind of integration of a noisy motor command instructing a change in force level and there is increasing evidence such integration – see for example the arguments made in Shadmehr, 2017, that even during isometric contractions, motor cortex (but not spinal) activity tends to be highly phasic. Alternatively, many other aspects of the true, complex neuromuscular dynamics may require state estimation of both forces and their first derivatives. To avoid getting bogged down in details, we have summarised our arguments in the text:

“The visual system can perceive relatively high frequencies (up to flicker-fusion frequencies above 10 Hz). However, while feedforward movements can in some cases approach these frequencies, discrepancies while tracking slowly moving objects in the physical world are unlikely to change quickly. The limbs and real-world targets will tend (to a first approximation) to move with a constant velocity unless acted upon by a force. Moreover, even for isometric tasks the drift in force production is dominated by low-frequency components (Baweja et al., 2009, Slifkin and Newell, 2000), possibly consistent with neural integration in the descending motor pathway (Shadmehr, 2017). Given inherent uncertainties in sensation, an optimal state estimator should attribute high-frequency errors to sensory noise (which is unconstrained by Newtonian and/or neuromuscular dynamics).

Formally, the task of distinguishing the true state of the world from uncertain, delayed measurements can be achieved by a Kalman filter, which continuously integrates new evidence with updated estimates of the current state, evolving according to a model of the external dynamics (Figure 3A). For simplicity we used Newtonian dynamics, although similar results would likely be obtained for other second-order state transition models.”

Reviewer #2 comment:In conveying and interpreting the above results, the authors employ control-theory style models. This is important: the models help to transcend hand-waviness and give concrete examples of how a system might work and what one would expect empirically. Certain results (e.g., a peak in frequency content, the shift in that peak with imposed delay, and the appearance of higher harmonics) are actually much easier to understand with a concrete model. Given this, I would not recommend removing the modeling…The most important aspects of the interpretation don't depend critically on the exact model – simply on there being some internal process with appropriate band-pass properties. The authors themselves note this in the Discussion. Thus, some balance needs to be struck between using a concrete model to illustrate a broader point, and not forcing the reader to dig deep to understand a specific model that is rather post-hoc and is only one of many plausible models… More of the focus should be on the general properties of the data and what they imply at a broad level. Control-theory-style models can (and probably should) be used to illustrate fundamental properties, but less should be made of specific pet models. This is especially true when the conceptual point is a fairly general one, and when many related models might show the key properties. This would also allow the manuscript to be more readable. The manuscript could better indicate which details really matter and which don't. Then the reader would know when to pause and really understand a point, and when to read on.

We agree that the model should serve solely to illustrate the main conceptual points of our study, and is not intended to be a full simulation. As described above and documented in the new Figure 5, a linear feedback controller without delay-dependent adaptation does not appear to fit our data. We feel that the Smith Predictor model we use represents the simplest model that captures the key features of our data, but we do not deny that more complex models could also fit the data. We now explain this in the text:

“While not simulating the full complexity of upper-limb control, our model was intended to illustrate the interplay between intrinsic and extrinsic dynamics during tracking. More sophisticated models would likely exhibit qualitatively similar behavior so long as they also incorporated extrinsic, delay-dependent feedback and intrinsic, delay-independent dynamics.”

As well as in the Discussion:

“We used Newtonian dynamics to construct a simple two-dimensional state transition model based on both the cursor-target discrepancy and its first derivative. While this undoubtedly neglects the true complexity of muscle and limb biomechanics, simulations based on this plausible first approximation reproduced both the amplitude response and phase delay to sinusoidal cursor perturbations in humans, and the population dynamics of LFP cycles in the monkey.”

Remaining comments:Reviewer #1 comment:2) The sleep bits seem completely out of place until it appears near the end of the discussion. I would seriously consider dropping its mention from earlier in the manuscript because no one will have any idea what is going on anyway.

We hope that our revised Introduction explains more clearly the importance of the observation that cortical dynamics are consistent across multiple behaviours and sleep. This provides evidence for an ‘intrinsic’ origin for these dynamics, as opposed to the alternative hypothesis that these dynamics reflect feedback loops from the environment (which are absent in sleep). We therefore included ‘sleep’ in our manuscript as it seems relevant to the present study, and may also provide inspiration to potential readers.

Reviewer #1 comment:4) The authors use the intervention of increasing visual delays and visual perturbations as a means for probing the state of the system. How does this limit the generalizability of their findings? There is, for example, the following paper that describes the tradeoff of using fast but crude somatosensory inputs versus slow but acute visual ones. Might be worth considering more deeply (477-481) the interplay between many modalities in terms of the OFC framework.http://www.jneurosci.org/content/jneuro/36/33/8598.full.pdf

Although not a focus of our study, the optimal state estimation framework we used in our modelling does indeed generalise to the integration of different sensory modalities. We have added reference to the Crevecoeur study and the issues involved in optimal state estimation to the discussion:

“We suggest that the computations involved in optimal tracking behaviors are likely distributed across multiple cortical areas including (but not limited to) M1, with local circuitry reflecting multiple dynamical models of the various sensory and efference copy signals that must be integrated for accurate control. These could include the estimation of the position of moving stimuli based on noisy visual inputs (Kwon et al., 2015), as well the optimal integration of visual and somatosensory information which may have different temporal delays (Crevecoeur et al., 2016).”

Reviewer #1 minor comment:1) The figures are very small, it was very hard to appreciate any details in these. They need to be substantially bigger in print.

Some of the figures were inadvertently resized when embedding into the text. We have substantially enlarged all figures in this resubmission.

Reviewer #2 comment:2) The term 'Optimal Control Model' is used often. I agree that the Kalman filter is an optimal estimator of the state. I'm not so sure about the whole model. I generally think of 'Optimal Feedback Control' (OFC) models as having a feedback law that is optimized based on a particular cost function. Typically that feedback law is time-varying and specific to each action (e.g., it is different when reaching right vs left). Perhaps the model presented is optimal in the case of a very simple cost function (keep the output near zero at all times). I'm not sure. I thus ask the authors to think carefully regarding whether the model is really an OFC model, or merely a good feedback controller that employs optimal state estimation.

For finite-horizon tasks (e.g. discrete reaches), optimal feedback laws have time-varying gains. However, for infinite-horizon tasks (e.g. continuous tracking), feedback laws converge to a steady-state solution with a constant gain (i.e. an LQR regulator). In this sense, our model is an Optimal Control Model (Kalman Filter + LQR regulator), albeit only under linear-quadratic-gaussian assumptions.

Reviewer #2 comment:3) I found Figure 1F confusing when first described. It isn't a feedback system, but is described as such. Only later do we learn that certain types of feedback control systems can be formally reduced to the diagram in 1F. This is one of those places where the cognitive load on the reader spikes.

We agree that this is slightly confusing, but feel that showing the feedback comb filter would be even more confusing since we later show that such feedback controller does not (easily) fit the force responses in our perturbation expt (Figure 5). Instead we have included a sentence of explanation:

“In signal processing terms, subtracting a delayed version from the original signal is known as comb filtering (Figure 1E). Although comb filters subtracting in either feedforward or feedback directions have qualitatively similar behavior, we illustrate only the feedforward architecture in Figure 1E as we will later show this to match better the experimental data.”

Reviewer #2 comment:4) Speaking of cognitive load, I initially found the third row of Figure 2 (panels C,D,E) rather confusing. It isn't immediately obvious exactly what is different from the above row. Even once one figures that out, the summary results in panel H seem not to obviously agree with the results in panels F and G. e.g., the peak at 2 Hz is much higher in panel F than G, yet this isn't reflected in panel H. I think I understand this in the end (after subtracting baseline and taking the square root, the difference is pretty small), but this threw me for a while.There are other instances where I had to struggle a bit to understand figure panels. In the end I was typically able to (and satisfied when I did) but I felt I shouldn't have had to work so hard.

We have modified the text to explain more clearly the differences between cursor and force responses, as well as the fact that these are not derived directly from the power spectra, but instead from the components of these responses that are phase-locked to the perturbations.

“Figure 2C,D and Figure 2—figure supplement 1 overlay cursor velocity spectra in the presence of each perturbation frequency (with feedback delays of 0 and 200 ms), again calculated over a 10 s window beginning 5 s after the trial start…”

“Figure 2E shows the amplitude response of cursor movements at each frequency for both delay conditions. Unlike a power spectrum, the cursor amplitude response measures only cursor movements that are phase-locked to the perturbation (normalized by the perturbation amplitude), and therefore estimates the overall transfer function of the closed-loop control system.”

“Figures 2F,G and Figure 2—figure supplement 2 show power spectra of the angular velocity derived from subject’s forces, under feedback delays of 0 and 200 ms. Note that this analysis differs from Figure 2C,D in that we now consider only the force generated by the subjects, rather than the resultant cursor movement (which combines these forces with the perturbation). Figure 2H shows the corresponding force amplitude response for each perturbation frequency. The force amplitude response measures only force responses that are phase-locked to the perturbation (normalized by the perturbation amplitude) and is related to the transfer function within the feedback loop.”

Reviewer #2 comment:5) The Kalman-based explanation for the band-pass property is a reasonable one, but there are other reasonable explanations as well. This is appropriately handled in the Discussion. Yet this broader perspective comes rather late. This caught my eye because the authors connect their results (both behavioral and LFP-based) with recent findings that show ~2 Hz quasi-oscillatory dynamics in motor cortex. That is indeed a plausible connection, but not obviously consistent with the Kalman-filter-based explanation. For brisk reaches, plotting cursor position vs cursor velocity would produce an oscillation much higher-frequency than 2 Hz. Instead, it is the muscle activity that has ~2 Hz features during reaching. This is perhaps consistent with the Kalman idea, but at that point the quantity being filtered would be muscle activity not cursor position. I don't think these facts detract from the authors' findings, but they do suggest that interpretation could benefit if it were less tied to a very specific model.

We did not train our monkeys to make especially brisk (or slow) movements so we cannot comment on the effect that would have on the cortical dynamics we observed. However, our understanding of Churchland et al., 2012, is that the frequency of cortical cycles is not greatly altered in this case. This is at least what our simple model would predict. Fast/slow movements could be accommodated in our model by increasing/decreasing the controller gain. Nonetheless, in our interpretation, the rotational dynamics arises in the Kalman filter which we assume to be independent of the controller. Note that the cycles reflect the state estimates, not the actual discrepancies which may under some circumstances change more abruptly. Moreover, since the state estimator drives corrective movements, these frequencies might then show up in muscle activity. Since we were modelling a continuous tracking task, excitation to the state estimator was broadband motor noise. However, in the case of a discrete reach, the input to the state estimator would be a sudden jump in target position. This would act as an impulse into the state estimator, but the resultant evolution would nonetheless be governed by the same dynamics and exhibit the same rotational structure as in the case of tracking. We have added this to the discussion:

“Instead we interpret these intrinsic dynamics as implementing a state estimator during continuous feedback control, driven by noise in motor and sensory signals. We used Newtonian dynamics to construct a simple two-dimensional state transition model based on both the cursor-target discrepancy and its first derivative. While this undoubtedly neglects the true complexity of muscle and limb biomechanics, simulations based on this plausible first approximation reproduced both the amplitude response and phase delay to sinusoidal cursor perturbations in humans, and the population dynamics of LFP cycles in the monkey. We suggest that for discrete, fast movements to static targets, transient cursor-target discrepancies effectively provide impulse excitation to the state estimator, generating a rotational cycle in the neural space. Note that this account also offers a natural explanation of why preparatory and movement-related activity lies along distinct state-space dimensions, since the static discrepancy present during preparation is encoded differently to the changing discrepancy that exists during movement. At the same time, the lawful relationship between discrepancy and its derivative couples these dimensions within the state estimator and is evident as consistent rotational dynamics across different tasks and behavioral states.”

Of course, it could be argued that with extensive training on fast reaches, the state estimator might be expected to adapt to filter less of the high-frequency state changes. We discuss a related point:

“It would be interesting in future to vary these noise characteristics experimentally (e.g. by artificially degrading visual acuity or by extensively training subjects to produce faster or more accurate movements) and examine the effect on perturbation responses.”

Additional data files and statistical comments:There are certainly a few places where formal statistical tests (most likely bootstraps) could be added. For example, some of the features in Figure 7AB are quite small but potentially very revealing. Backing up their presence with statistical tests is thus appropriate.

Statistical tests have now been added to the (renumbered) Figure 8 and Figure 8—figure supplement 1.

[Editors' note: further revisions were suggested, as described below.]

[…] The first submission of this manuscript was a bit of a tough read (all reviewers were of this opinion). We find the revised version much improved. One reviewer does offer some minor revisions. His comments follow:"I really only have comments regarding changes to improve readability… Even though a couple of my comments are lengthy, I have labelled all these comments as 'minor', as I feel confident they can be readily addressed. I trust the authors to digest these points and to figure out what changes should be made.Minor comments:The authors use the new Figure 5 to address a criticism raised in the last round. While their model does a nice job of accounting for the data, it makes the potentially implausible assumption that the internal delay (in the Smith predictor) is able to rapidly update itself to reflect the sum of internal and imposed delays. The concern is that, because delays are different on different trials, it seems implausible that internal delay would always match the imposed delay. To address this concern, the authors point out that the brain has more opportunity to adapt than we had initially supposed: subjects were able to 'get used' to the delay for 5 seconds before the section of the data being analyzed. Furthermore, the new Figure 5 demonstrates that some kind of adaptation does occur during this time: models fit to the data need different parameters for the first 5 seconds versus the last 5.I still have my doubts that 5 seconds is enough time to recalibrate an internal estimate of the feedback delay. That said, other aspects of the model are very successful, and in general, I think the results go beyond any particular model. The paper would still be interesting and the take-home messages similar even if a somewhat different feedback model had to be proposed. For these reasons, I am largely satisfied, at a scientific level, with the way in which this concern has been addressed. That said, I found the newly added section to be confusing in multiple ways.First, the way the section is introduced doesn't help the reader understand why the section (and the large accompanying figure) is there. The sentence beginning with 'However, since…' is helpful only if you already have thought about this problem. I would suggest that the section begin by confronting the problem head-on, and clearly stating that a challenge for the model is that it would have to adapt very rapidly. Therefore it is worth asking whether some form of rapid adaptation occurs. Fully understanding the motivating concern would help the reader understand why this section is here (and allow them to make up their own mind regarding how convinced they are).Other aspects of this new section were also confusing to me. Subsection “Emergence of delay-specific predictive control during individual trials” states 'We asked which model could better explain the experimentally-observed force amplitude responses by inferring the associated intrinsic dynamics under both architectures.' This left me expecting more plots like those in Figure 4. But the analyses in Figure 5 don't (I think) really address this issue. Rather, they show that some kind of adaptation occurs. Given that it does, we have less reason to doubt a central modeling assumption (after all, any model would have to adapt in some way). This seems a reasonable argument, but that isn't how the section was motivated or set up.In this context, I also found the plots in E and F to be a bit confusing. They are plots of the gains used to fit the data, not plots of gains produced by the model. The plots in F seem to be a better match to the data, which at first seems like the point of the plots. However, I think that is actually a red herring. The main point (I think) is just that some kind of adaptation has to be assumed, which addresses the concern that fast adaptation is implausible.This section should be carefully rewritten to be more explicit regarding why the analysis was performed, how to interpret the plots, and what exactly is being concluded.

We have reworded this section to clarify that our analysis in fact demonstrates three points: (1) there is fast adaptation during the course of a trial, (2) this adaptation is specific to different delay conditions, and (3) adaptation results in behaviour consistent with an appropriately calibrated internal Smith Predictor loop. This third point is important and justifies our use of a Smith Predictor to model our main results. We do not agree that ‘different feedback models’ (i.e. without a Smith Predictor loop) would equally well explain these results, since any such model would need to exhibit the strange delay-dependent gain peaks seen in Figure 5E. Moreover, these would need to emerge over exactly the same fast time-course that troubles the reviewer! Finally, it would be difficult to explain why such delay-dependent gain peaks should emerge in the first place. By contrast, the Smith Predictor model does not require delay-dependent gain peaks (Figure 5F), is consistent with our data, and is amply justified from a theoretical perspective as an effective strategy for dealing with feedback delays. However, we agree that it is up to readers to decide how convinced they are.

For Figure 1E, where did lines come from? Regression? This seems to be the case (it is implied by the legend of Table 1) but the actual figure legend doesn't explicitly state this and I don't think the Results do either.

These are indeed regression lines and this is now stated in the figure legend.

Along similar lines, why not add the lines predicted by the model to Figure 1E? The info is currently presented in Table 1, but would likely be more than compelling if added to this key plot. It would be nice to see that the slopes and intercepts match fairly well (it needn't be perfect of course).

This is perhaps a philosophical point, but we prefer the approach of deriving confidence intervals for key metrics (slopes/intercepts) directly from the data, and then showing that these are consistent with a particular model. The approach suggested by the reviewer may be equally valid but would not provide as informative a description of the data.

I liked the “Outline” section but the heading “Outline” is perhaps a bit heavy handed?

We have renamed this section “Overview”.

My impression was that the cursor is not constrained to live on the circle. E.g., there could be radial errors (unanalyzed here). If so that should be stated explicitly in the Materials and methods (my apologies if I missed this).

This is correct and now stated explicitly in the Materials and methods:

“Although the 2D cursor position was not constrained to follow the target trajectory, we did not analyze off-trajectory deviations. For simplicity, kinematic analyses were based on the time-varying angular velocity of the cursor subtended at the center of the screen”

Subsection “Intrinsic dynamics in the visuomotor feedback loop” states 'Analyzed in this way, amplitude responses were largely independent of extrinsic delay.' When I first read this, I found it all rather confusing. Analyzing different ways gives different results? Which is correct? Is there some sort of conflict? It all makes sense in the end, but a bit of help at this critical juncture might save someone some temporary confusion.

We have reworded this section:

“Unlike the cursor amplitude responses described previously, force amplitude responses were largely independent of extrinsic delay.”

Regarding Figure 4J. This plot has the potential to generate some confusion, as it is actually data, but presented in the context of a figure where nearly all the other panels (A-I) contain only model behavior. Also, there is a dashed line that seems like it might be a model prediction, but might in fact just be the result of a linear regression (this isn't state). This all needs to be clarified. Also, if this is an analysis of data testing a model prediction, then the model prediction should be shown. Otherwise we are comparing to a prediction that isn't illustrated.

We have replaced the dashed line with the model prediction (which is the same as the phase delay shown in Figure 4I). Note that in altering this figure, we realised a slight inaccuracy in our previous figure. Previously we had averaged the frequencies of peaks seen in individual subject velocity spectra. We now use the frequencies of the peaks seen in the average velocity spectra (i.e. Figure 1D), as was stated in the legend. This gives a slightly better fit to the model, although the overall pattern is unchanged.

Subsection “Intrinsic cortical dynamics are unaffected by feedback delays” paragraph four: 'the two neural population<s>'

This has been corrected.